# *Patronus*: INTERPRETABLE DIFFUSION MODELS WITH PROTOTYPES

**Nina Weng, Aasa Feragen, Siavash Bigdeli**
Technical University of Denmark
`{ninwe, afhar, sarbi}@dtu.dk`

## ABSTRACT

Uncovering the opacity of diffusion-based generative models is urgently needed, as their applications continue to expand while their underlying procedures largely remain a black box. With a critical question – how can the diffusion generation process be interpreted and understood? – we proposed *Patronus*, an interpretable diffusion model that incorporates a prototypical network to encode semantics in visual patches, revealing *what* visual patterns are modeled and *where* and *when* they emerge throughout denoising. This interpretability of *Patronus* provides deeper insights into the generative mechanism, enabling the detection of shortcut learning via unwanted correlations and the tracing of semantic emergence across timesteps. We evaluate *Patronus* on four natural image datasets and one medical imaging dataset, demonstrating both faithful interpretability and strong generative performance. With this work, we open new avenues for understanding and steering diffusion models through prototype-based interpretability.
Our code is available at nina-weng.github.io/patronus.github.io.

## 1 INTRODUCTION

The generative capabilities of modern machine learning models, particularly diffusion models, have advanced significantly, enabling the creation of highly realistic samples that closely resemble real-world data. However, their opacity raises critical concerns, including bias amplification (Luccioni et al., 2023), unsafe content (Qu et al., 2023), and copyright violations (Vyas et al., 2023). Their lack of transparency makes it difficult to detect and mitigate these risks, highlighting a fundamental question: *How can the diffusion generation process be interpreted and understood?* Specifically, **what** visual patterns emerge during generation, **where** and **when** they appear, and to what extent they can be **controlled**. Addressing these questions is essential, not only for improving generative ability but also for ensuring interpretability, transparency, and ethical deployment, aligning with regulatory frameworks such as the EU AI Act.

Existing approaches to improving interpretability in diffusion-based visual generation typically fall into two categories. The first relies on post-hoc analysis to investigate how semantic information are encoded in intermediate representations (Kwon et al., 2022; Lee et al., 2023; Park et al., 2023; Haas et al., 2024). However, this method is inherently retrospective and limited in its ability to provide direct control over generation. The second approach introduces additional encoder-based semantic vectors for diffusion guidance (Preechakul et al., 2022; Leng et al., 2023; Wang et al., 2023b), which improves controllability, but often resulting in representations that are difficult to interpret. Moreover, these methods tend to capture global (e.g. face shape, pose) rather than fine-grained patterns (e.g. hair/make-up details, facial expressions), and the latter are crucial for interpretability.

To address these limitations, we aim to develop a method that fulfills two key objectives: (1) embedding interpretability directly into the model architecture, thereby providing **intrinsic transparency** and eliminating the need for post-hoc analysis of high-dimensional latent features; (2) moving interpretability a step further from global to **local semantic meanings** and enabling the controllability.

In this work, we propose *Patronus*, an interpretable diffusion-based generative model integrated with a prototypical network for semantic encoding (Fig. 1). Our approach trains prototype features to capture localized patterns within image patches, and uses prototype activation vector to encode the presence of semantic information for diffusion guidance. This design effectively reduces the

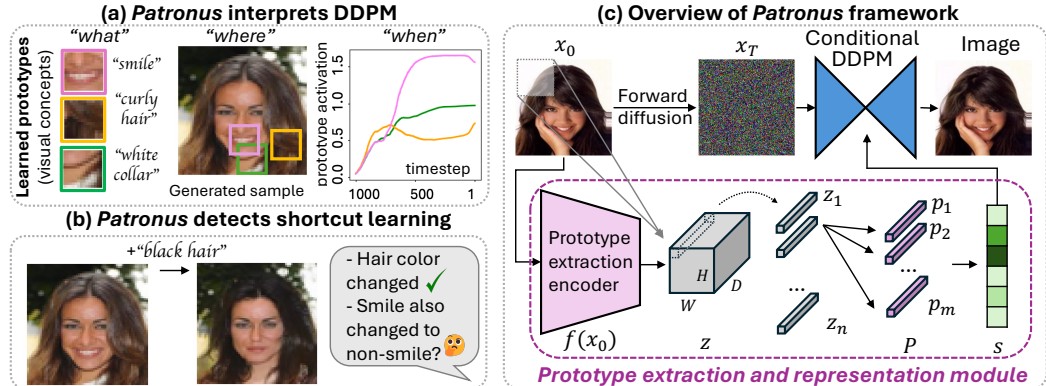

Figure 1: **Proposed *Patronus* model**. **(a) Interpretability**: By integrating a prototypical network as the encoder, *Patronus* learns semantic prototypes (*"what"*) and explains the generative process by revealing *where* and *when* they emerge. **(b) Diagnosis:** Patronus could detect unwanted correlation (e.g., in this case hair color and smile) learned from training data. **(c) Overview of *Patronus*:** contains a prototypical network for prototype extraction and a conditional DDPM for generation.

latent dimensionality while preserving sufficient semantic information. *Patronus* allows directly visualizing the semantics of the learned prototypes by manipulating the condition signals on specific prototypes, thereby enhancing interpretability and enabling the diagnosis of shortcut learning. Beyond that, *Patronus* further reveals when prototypes emerge during the denoising process, providing temporal insights and guiding more efficient editing.

**Our contributions are summarized as follows:**

- We propose *Patronus*: An interpretable diffusion model for image generation which incorporates a prototypical network for prototype learning and representation, alongside a conditional guidance by a prototype activation vector – entirely without extra annotations.
- We introduce a novel method for visualizing learned prototypes, revealing the semantic meanings of them and how they engaged in the generation process.
- Through extensive experiments, we show that *Patronus* effectively captures semantically meaningful features within images, and achieves competitive latent quality and generation quality compared to SOTA methods.
- We empirically demonstrate the potential of *Patronus* to diagnose hidden bias by detecting shortcut learning, offering a valuable tool for mitigating biases in generative models and promoting fairness in their deployment.

## 2 RELATED WORK

### 2.1 PROTOTYPE-BASED INTERPRETABILITY

Our work is closely related to and motivated by ProtoPNet (Chen et al., 2019), which builds interpretable deep learning frameworks by learning *prototypes*, intermediate representations of visually similar patterns between training and inference images.

Under such a design, a key challenge is visualizing the learned prototypes. Li et al. (2018) used a decoder to reconstruct prototypes, which resulted in blurry visualizations due to data ambiguity. ProtoPNet instead identified the closest matching encoded representations from the training set via distance comparisons. However, this approach cannot guarantee capturing the true representative mode of the distribution. Follow-up works (Donnelly et al., 2022; Wang et al., 2023a; Ming et al., 2019; Ghosal & Abbasi-Asl, 2021) explore alternative prototype design but *do not* enhance visualization. We propose a new method for prototype visualization that allow finding the most probable visual representation of a prototype. Additionally, the aforementioned studies only focus on *classification*, our work focus on learning and visualizing prototypes for *generative* models.

## 2.2 INTERPRETABLE DIFFUSION MODELS

Recent work on explaining diffusion models has focused on the following two directions.

**Semantic Interpretation of Internal Features.** Diffusion models were traditionally viewed as lacking internal representations compared to other generative models, like VAE (Kingma & Welling, 2013) and GAN (Goodfellow et al., 2014). However, Kwon et al. (2022) pointed out that *diffusion models possess a semantic latent space* within the U-Net's intermediate layer. They leverage the bottleneck for semantic control over the denoising process and propose the asymmetric reverse process to edit the image based on the discovered semantic meaning.

Building on this idea, researchers have sought to interpret the latent space of diffusion models by grouping the latent vectors with different noise schedules (Lee et al., 2023), applying pullback metrics to obtain meaningful local latent basis (Park et al., 2023), and uncovering semantically meaningful directions by both PCA and linear properties of the semantic latent space (Haas et al., 2024). Other works (Si et al., 2024; Tumanyan et al., 2023) extract structural information from the latent space and use it for I2I translation or prompted generation tasks.

This line of work focuses on post-hoc analyses of existing diffusion models, whereas our approach provides *intrinsic* interpretability by design. Moreover, while prior studies often rely on annotated data or external classifiers to uncover the semantic latent space, *Patronus* learns prototypes in a fully *unsupervised* manner.

**Autoencoder-based semantic feature extraction for guidance.** An alternative way to enhance the interpretability of diffusion models is to integrate an additional encoder to extract semantic features for guidance. DiffAE (Preechakul et al., 2022) pioneered this idea using a learnable encoder for latent semantics. Expanding on this, DiffuseGAE (Leng et al., 2023) proposed a group-supervised AutoEncoder to achieve better latent disentanglement, while InfoDiffusion (Wang et al., 2023b) reduced latent dimensionality and enforced mutual information constraints for more effective learning.

While these methods extract global semantic features, *Patronus* instead captures local features through a prototypical network. Another key distinction lies in how the diffusion model is guided: Rather than direct semantic information, we use the *prototype activation vector*. This approach significantly reduces the dimensionality needed for diffusion guidance while preserving enough capacity in prototype features to encode the same semantic information.

## 3 PATRONUS: INTERPRETABLE DIFFUSION MODEL

Our proposed model, **Patronus: Prototype-Assisted Transparent Diffusion Model**, is illustrated in Fig. 1c. Designed to enhance transparency and interpretability in diffusion models, *Patronus* incorporates a prototype extraction and representation module (Fig. 1c-bottom). This module learns patch-based prototypes within the image and computes similarity score for each prototype, which are then used to condition the diffusion process (Fig. 1c-top). We elaborate on the details of the prototype extraction and representation module in Sec. 3.1 and the conditional DDPM in Sec. 3.2. Furthermore, we explain how transparency and interpretability are achieved, including sampling strategy and manipulations, in Sec. 3.3. We detail the unconditional sampling strategies in Sec. 3.4, and furthermore prove that adding conditions does not degrade the denoiser training in Sec. 3.5.

### 3.1 PROTOTYPE EXTRACTION AND REPRESENTATION

This module, inspired by ProtoPNet, consists of two key components: The **prototype encoder** transforms input images into patch-based feature representations. Utilizing the properties of convolutional neural networks (CNNs), each output neuron in the feature map corresponds to a specific patch of the input image, determined by the network's receptive field. This patch-based representation enables the model to focus on localized patterns and learn fine-grained prototypes. The **activation vector** is derived by calculating similarity scores for each learned prototype, based on the distance between encoded patches and prototypes, where higher scores indicate stronger matches.

This module works as follows: As shown in Fig. 1c, given an input image $x_0$, the prototype encoder $f$ extracts features $z = f(x_0)$ into a tensor of shape $H \times W \times D$. In this work, the encoder is a

4-layer Conv–ReLU encoder. The network learns $m$ prototypes in the latent feature space during training, denoted as $P = \{p_j\}_{j=1}^m$, each with the shape $1 \times 1 \times D^1$. Each prototype $p_j$ can be interpreted as a latent encoding of a patch in the original pixel space. This patch, importantly, need not exist in the dataset but should lie within the plausible data distribution.

For the encoder output $z = f(x_0)$, each spatial region within $z$ that corresponds to the same size as a prototype ($1 \times 1 \times D$) can be interpreted as representing a patch of $x_0$. Thus, $z$ can be decomposed into smaller regions as follows: $z = \{z_i\}_{i=0}^n$, where $n$ denotes the total number of patches encoded in $z$. In our case, $n = H \times W$.

To calculate the similarity between the encoded features $z$ and the learned prototypes $P$, we begin by computing the squared $L2$ distance between each spatial feature $z_i$ and each prototype $p_j$: $d^2(z_i, p_j) = ||z_i - p_j||^2$. Next, the minimum distance across the spatial dimensions is selected for each prototype: $d^2_{min,j} = max(-d^2_j,$ kernel size $= (H, W))$, where $d^2_j$ is the set of distances for the $j_{th}$ prototype across all spatial positions, and the kernel size aligns with the feature map's spatial dimensions $H$ and $W$. The sequence of minimum distances for all prototypes is converted into an activation vector $s$ using a log transformation: $s = \log(\frac{d^2+1}{d^2+\epsilon})$ , where $\epsilon$ is a small positive constant.

## 3.2 CONDITIONAL DIFFUSION PROCESS

Denoising diffusion probabilistic models (DDPM) form a class of generative models that learn data distributions by iteratively denoising a noisy latent representation. The process involves **(1) a forward diffusion process**, where Gaussian noise is progressively added to a data sample $x_0$ over $T$ timesteps, producing noisy latents $x_t$, defined as

$$q(x_t \mid x_{t-1}) = \mathcal{N}(x_t; \sqrt{\alpha_t} x_{t-1}, (1 - \alpha_t)\mathbf{I}). \tag{1}$$

Here, the marginal distribution of $x_t$ given $x_0$ is:

$$q(x_t \mid x_0) = \mathcal{N}(x_t; \sqrt{\bar{\alpha}_t} x_0, (1 - \bar{\alpha}_t)\mathbf{I}), \tag{2}$$

where $\bar{\alpha}_t = \prod_{i=1}^t \alpha_i$ and $\alpha_t = 1 - \beta_t$, where $\beta_t$ is the variance of the Gaussian noise added at $t$. Furthermore, DDPMs rely on **(2) a reverse generative process** that removes noise given $t$:

$$p_\theta(x_{t-1} \mid x_t) = \mathcal{N}(x_{t-1}; \mu_\theta(x_t, t), \Sigma_\theta(x_t, t)). \tag{3}$$

To enable **conditional generation**, we modify the reverse process to be conditioned on the prototype activation vector $s$. Therefore the updated reverse process is:

$$p_\theta(x_{t-1} \mid x_t, s) = \mathcal{N}(x_{t-1}; \mu_\theta(x_t, t, s), \Sigma_\theta(x_t, t)). \tag{4}$$

The training objective remains based on the standard noise-prediction loss used in DDPM. For a given noisy sample $x_t$, timestep $t$ and noise $\epsilon$ the model minimizes the loss:

$$\mathcal{L}_{ddpm} = \mathbb{E}_{x_0, \theta, t}[||\epsilon - \epsilon_\theta(x_t, t, s)||^2], \tag{5}$$

where $\epsilon_\theta$ is the learned denoiser. As the loss indicates, our guidance does not change the model's output – it only encourages its reasoning to utilize prototypes for transparency.

## 3.3 TRANSPARENCY AND INTERPRETABILITY OF PATRONUS

The similarity score $s_j$ quantifies the activation of the $j_{th}$ prototype in a given input, indicating the presence of specific semantic patterns. The model thus conditionally generates samples guided by interpretable semantic information.

**Visualizing learned prototypes.** Integrating a prototypical network as a semantic meaning extraction module brings inherent interpretability: Each learned prototype vector $p_j$ represents a patch in the image domain. In ProtoPNet, those patches are retrieved by greedily searching all candidate patches in the training set for the closest embeddings.

We argue that the learned prototypes do not need to correspond directly to specific training patches but should instead align with the overall distribution of the training data. To support this, we propose a novel prototype visualization method with the following steps:

---

[1] whose generalization to $H_1 \times W_1 \times D$ as in Chen et al. (2019) is straightforward.

1. Compute the activation vector $s = \{s_j\}_{j=1}^m$, for a given sample $x_0$, where $s_j$ represents the similarity score between $x_0$ and the $j_{th}$ prototype.

2. For target prototype $J$, increase its similarity score $s_J$ to the plausible maximum while keeping all other scores unchanged. The updated activation vector $s' = \{s'_j\}_{j=1}^m$ is defined as: $s'_j = \begin{cases} s_j, & \text{if } j \neq J \\ max(s_X), & \text{if } j = J \end{cases}$. Here, $s_X$ represents similarity scores from a representative subset, constraining $s_J$ within a plausible range.

3. Using the updated activation vector $s'$ to sample a new image $x'$ conditioned on $s'$.

4. Identify the most activated patch $x'_i$ in $x'$ that corresponds to the target prototype $J$. This patch $x'_i$ serves as the visual representation of $p_J$.

This method could also be used to visualize prototypes in other prototypical deep learning models.

**Manipulation using the prototype activation vector.** Manipulating images is a natural downstream task for *Patronus*, as adjusting a specific prototype similarity score $s_j$ and conditionally generating a new sample allows us to effectively and semantically control the image content:

$$p_\theta(x_{t-1} \mid x_t, s') = \mathcal{N}(x_{t-1}; \mu_\theta(x_t, t, s'), \Sigma_\theta(x_t, t)) \tag{6}$$

**Deterministic reverse process via DDIM sampling.** Both the visualization of prototypes and their manipulation via activation vectors build on the DDPM sampling process. However, for stricter control over the stochasticity introduced by random noise, the Denoising Diffusion Implicit Models (DDIM) sampler provides an alternative approach. Here, the denoiser is given by $p_\theta(x_{t-1} \mid x_t, s) = \mathcal{N}(x_{t-1}; \mu_t, \sigma^2 \mathbf{I})$, with mean $\mu_t = \sqrt{\bar{\alpha}_{t-1}} \cdot \hat{x}_0 + \sqrt{1 - \bar{\alpha}_{t-1} - \sigma^2} \cdot \epsilon_\theta(x_t, t, s)$ and variance $\sigma^2 = \eta^2 \cdot \frac{1 - \bar{\alpha}_{t-1}}{1 - \bar{\alpha}_t} \cdot (1 - \frac{\bar{\alpha}_t}{\bar{\alpha}_{t-1}})$. By setting $\eta = 0.0$, the process becomes deterministic, leading to the following update for the reverse process: $x_{t-1} = \sqrt{\bar{\alpha}_{t-1}} \cdot \hat{x}_0 + \sqrt{1 - \bar{\alpha}_{t-1}} \cdot \epsilon_\theta(x_t, t, s)$. Here, $\hat{x}_0$ is the estimated denoised image, computed as $\hat{x}_0 = \frac{1}{\sqrt{\bar{\alpha}_t}}(x_t - \sqrt{1 - \bar{\alpha}_t} \cdot \epsilon_\theta(x_t, t, s))$.

This DDIM sampling requires $x_T$, which represents the initial noise in the diffusion process. This could be obtained by performing a deterministic backward generative process, given by $x_{t+1} = \sqrt{\bar{\alpha}_{t+1}} \cdot \hat{x}_0 + \sqrt{1 - \bar{\alpha}_{t+1}} \cdot \epsilon_\theta(x_t, t, s)$.

### 3.4 UNCONDITIONAL SAMPLING STRATEGY

For unconditional sampling, we train an auxiliary latent diffusion model $p(s_{t-1}|s_t, t)$ to sample $s$. During training, we first jointly optimize the prototypical encoder with the conditional DDPM; subsequently, the latent diffusion model is trained with the parameters of prototype encoder frozen.

### 3.5 ADDING THE CONDITION TO THE OBJECTIVE

In *Patronus*, the prototype encoder is jointly optimized with the denoiser. To show that this simultaneous training does not degrade the generated distribution, we analyze how updating the condition $s$ affects the likelihood. For ease of derivation, we use the Evidence Lower Bound (ELBO) as an equivalent objective to generalize denoising losses (Ho et al., 2020).

**Proposition.** *Any ELBO-improving update for the encoder always leads to progress. Such an update either increases the conditional likelihood of the data under the model or reduces the KL divergence between the generated distribution and the underlying data distribution.*

*Proof.* Let $z = x_{1:T}$ denote the forward noised latents generated from a fixed forward process $q(z \mid x)$, as is standard in diffusion models. For parameters $s$, define the per-sample evidence lower bound (ELBO)

$$\text{ELBO}(x; s) = \mathbb{E}_{q(z|x)}[\log p_\theta(x, z \mid s) - \log q(z \mid x)],$$

and the conditional log-likelihood

$$\log p_\theta(x \mid s) = \text{ELBO}(x; s) + \text{KL}\big(q(z \mid x) \,\big\|\, p_\theta(z \mid x, s)\big).$$

Consider an update $s^i \mapsto s^{i+1}$ that satisfies $\text{ELBO}(x; s^{i+1}) \geq \text{ELBO}(x; s^i)$ for the data point $x$. Then the following identity holds:

$$\log p_\theta(x \mid s^{i+1}) - \log p_\theta(x \mid s^i) = \big[\text{ELBO}(x; s^{i+1}) - \text{ELBO}(x; s^i)\big] + \Delta\text{KL}(x), \qquad (7)$$

where $\Delta\text{KL}(x) = \text{KL}\big(q(z \mid x) \,\|\, p_\theta(z \mid x, s^{i+1})\big) - \text{KL}\big(q(z \mid x) \,\|\, p_\theta(z \mid x, s^i)\big)$.

Given that the ELBO difference term is positive, Equation 7 implies that one of the following (improving) outcomes must occur:

**Conditional likelihood improves.** If $\Delta\text{KL}(x) \leq 0$, then

$$\text{KL}\big(q(z \mid x) \,\|\, p_\theta(z \mid x, s^{i+1})\big) <= \text{KL}\big(q(z \mid x) \,\|\, p_\theta(z \mid x, s^i)\big).$$

meaning that the model posterior $p_\theta(z \mid x, s^{i+1})$ more closely matches the forward distribution $q(z \mid x)$. This reduces the gap between the ELBO and the true conditional log-likelihood, yielding a tighter variational bound.

**Generated samples become more probable.** If $\Delta\text{KL}(x) > 0$ and considering that ELBO is also increased, then necessarily

$$\log p_\theta(x \mid s^{i+1}) \geq \log p_\theta(x \mid s^i),$$

i.e. the model assigns higher probability to $x$.

Hence, any update of the encoder that increases the ELBO either increases the conditional likelihood of the data or reduces the KL divergence between the model posterior and the forward process, guaranteeing progress in approximating the data distribution. $\qquad\square$

This analysis is inapplicable under alternative training objectives for latent condition (e.g., InfoDiff).

## 4 EXPERIMENTS AND RESULTS

In our experiments, we first assess the semantic meaning of learned prototypes (Sec. 4.1) through reconstruction, interpolation, extrapolation, and manipulation tasks. We also investigate prototype visualization and their consistency. We then analyze prototype quality and generation performance (Sec. 4.2, 4.3). Finally, we explore how *Patronus* aids in diagnosing shortcut learning in diffusion models by identifying potentially unwanted correlations via prototype similarity scores (Sec. 4.4).

We use five datasets: Quantitative evaluation on FMNIST (Xiao et al., 2017), CIFAR-10 (Krizhevsky et al., 2009), FFHQ (Karras, 2019); qualitative analysis on CheXpert (Irvin et al., 2019); and in-depth quantitative and qualitative experiments on CelebA (Liu et al., 2015). We use 100 prototypes for all datasets except FMNIST, which has 30. Prototypes are encoded with a shape of (1,1,128).

### 4.1 SEMANTIC MEANING OF THE LEARNED PROTOTYPES.

We firstly verify that the learned prototypes are *semantically meaningful* by the following analyses.

**Reconstructing image semantics from prototype activations.** We extract the prototype activation vector $s = Enc(x_0)$ and sample random noise $x_T \sim \mathcal{N}(0, \mathbf{I})$ to generate new images $\hat{x}(s, x_T)$. Fig. 2a presents one reconstructed image (using the same $x_T$) with three variations (using random $x_T$). The results show that the majority of semantic information in the images is accurately recreated, confirming that the prototypes effectively capture meaningful semantic features.

**Interpolation.** Given two images $x_0^1$ and $x_0^2$, we first retrieve its corresponding prototype activation vector and starting noise by reverse DDIM process: $(s^1, x_T^1)$ and $(s^2, x_T^2)$, and then generate new samples using $(\text{Lerp}(s^1, s^2; t), \text{Slerp}(x_T^1, x_T^2; t))$ for steps $t \in [0, 1]$, where Lerp/Slerp represents linear/spherical linear interpolation respectively. Results are shown in Fig. 2c.

**Visualization of the learned prototypes and consistency validation.** Unlike other autoencoder-based diffusion models, our approach is explicitly designed to yield an interpretable semantic latent space, where each prototype is trained to capture distinct semantic content. In practice, we realize this by amplifying a prototype's activation and extracting the most responsive patch (see Sec. 3.3). Fig. 3a shows selected prototypes with their semantic visualizations, and Fig. 3b demonstrates that the most activated patches remain semantically consistent across samples.

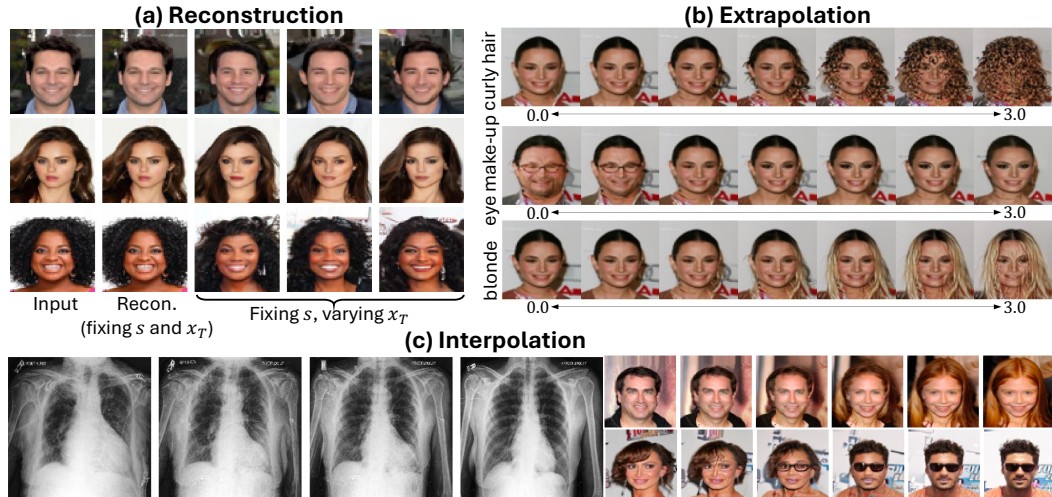

Figure 2: **Assessing the semantic meaning of the learned prototypes.** (a) Reconstruction. (b) Extrapolation. (c) Interpolation. Left: CheXpert, from 75-year-old female $w/o$ enlarged heart (left) to 27-year-old male $w/$ enlarged heart (right). Right: 2 examples from CelebA.

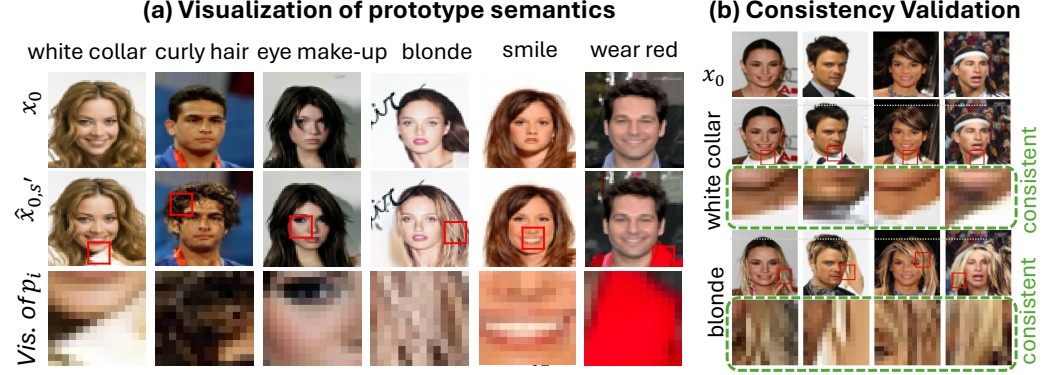

Figure 3: **Prototype visualization and consistency.** (a) Prototype visualization. Here, $x_0$ denotes the original image, $\hat{x}_{0,s'}$ denotes the generated image guided by condition $s'$, where $s'$ is the enhanced prototype activation on $j$-th prototype. Red square highlights the most activated patch, which is considered as the visual representation of the chosen prototype, also shown in the third row. Note that prototype semantics are not pre-annotated but inferred through observation. (b) Visualization across random samples demonstrates that each prototype encodes consistent semantics.

**Manipulation and extrapolation.** By adjusting the condition $s$ we can edit the image with specific semantic requests. Furthermore, pushing a selected dimension of $s$ to extreme values (ranging from 0.0 to 3.0) results in a smooth and continuous enhancement of the associated semantic information, as demonstrated in Fig.2b. Unlike interpolation, which remains within the observed range, this process extends beyond the original data distribution, making it an *extrapolation*.

## 4.2 PROTOTYPE QUALITY

**Prototype capability in semantic representation.** We test the prototype quality via a downstream classification task on the prototype activation vectors $s$ using a logistic regression classifier trained with 5-fold cross validation, reporting AUROC in Tab. 1 & 2. Our model outperform 3 out of 4 datasets in latent (prototype) quality, with particularly strong performance on CelebA and FFHQ. The lower latent quality for FMNIST may stem from *Patronus* prioritizing localized features, while DiffAE and InfoDiff emphasize global structures, which better capture the semantic information of

Table 1: **Prototype quality and generation quality on CelebA.**

|  | TAD ↑ | Attrs ↑ | Latent AUROC↑ | FID ↓ |
|---|---|---|---|---|
| DiffAE | 0.16±0.01 | 2.0±0.0 | 0.80±0.00 | 22.7±2.1 |
| InfoDiff | 0.30± 0.01 | 3.0±0.0 | 0.84±0.00 | 23.6±1.3 |
| $w/$ learned $z$ | 0.30± 0.01 | 3.0±0.0 | 0.84±0.00 | 22.3±1.2 |
| Patronus | **0.43±0.02** | **9.0±0.0** | **0.87±0.00** | 14.6±0.1 |
| $w/$ learned $s$ | **0.43±0.02** | **9.0±0.0** | **0.87±0.00** | **4.8±0.0** |

Table 2: **Prototype quality and generation quality on FashionMNIST, CIFAR-10, and FFHQ.**

|  | FMNIST | | CIFAR-10 | | FFHQ | |
|---|---|---|---|---|---|---|
|  | Latent AUROC ↑ | FID ↓ | Latent AUROC ↑ | FID ↓ | Latent AUROC ↑ | FID ↓ |
| DiffAE | 0.84±0.00 | 8.2±0.3 | 0.40±0.01 | 32.1±1.1 | 0.61±0.00 | 31.6±1.2 |
| InfoDiff | **0.84±0.00** | 8.5±0.3 | 0.41±0.00 | 32.7±1.2 | 0.61±0.00 | 31.2±1.6 |
| $w/$ learned $z$ | **0.84±0.00** | 7.4±0.2 | 0.41±0.00 | 31.5±1.8 | 0.61±0.00 | 30.9±2.5 |
| Patronus | 0.82±0.00 | 14.7± 0.3 | **0.54±0.01** | 32.9 ± 0.4 | **0.92±0.00** | 37.3±0.2 |
| $w/$ learned $s$ | 0.82±0.00 | **2.6 ± 0.1** | **0.54±0.01** | **8.0 ± 0.1** | **0.92±0.00** | **24.1 ± 0.1** |

FMNIST due to its high inter-class variability. For the datasets where semantic information is more localized, *Patronus* achieves a marked improvement.

**Prototype disentanglement.**    We quantify the disentanglement on CelebA using TAD (Yeats et al., 2022). Following Wang et al. (2023b), we first remove the highly correlated attributes, then compute the AUROC score for each dimension of the prototype activation vector $s$. An attribute is "captured" if any dimension achieves AUROC $> 0.75$. TAD is the sum of AUROC differences between the top two predictive dimensions per captured attribute. As shown in Tab. 1, *Patronus* outperforms previous models in both TAD and captured attributes.

### 4.3 GENERATION QUALITY

We assess generative quality using the Fréchet Inception Distance (FID), averaged over five random test sets of 10,000 images. Evaluation covers both unconditional and prototype-conditioned generation, where the latter incorporates learned prototype activation vectors from the test set (see Tab.1 & 2). *Patronus* significantly outperforms previous methods in prototype-conditioned generation across all four datasets. In the unconditional setting, it achieves state-of-the-art performance on CelebA and remains competitive on others; noting that its effectiveness depends on the training quality of both *Patronus* and the latent diffusion model.

### 4.4 DIAGNOSING SHORTCUT LEARNING IN DIFFUSION MODELS

We manipulated a subset of CelebA to introduce an unwanted correlation between hair color and smile: all blonde/brown-haired images smile, while black-haired ones do not. Results confirm the model captures the introduced bias, as increasing the black hair prototype shifts the smile property from "smile" to "non-smile" and vice versa (see Fig. 4a). Similar patterns emerge without subset manipulation. In Fig. 2b, decreasing "eye makeup" reduces female features. This emphasizes how *Patronus* can be utilized to discover unwanted model behavior, such as shortcut learning in this case.

## 5 DISCUSSION AND CONCLUSION

**Prototypes Emerge at Different Times during Generation.**    Given a generated image, *Patronus* reveals *when* each prototype emerges in the generation process by obtaining the prototype similarity score from estimated $\hat{x}_0$ at each timestep (more details in Appendix), as shown in the top-right corner of Fig. 1a. Thus, by subtracting the two time-sequential $s$ of the semantically-enhanced image $\hat{x}_{0,s\prime}$ and $x_0$, we illustrate how each prototype emerges temporally in the diffusion process. As shown in Fig. 4b, none of the prototypes have a significant emergence in the first 200 stages of generation. Interestingly, prototypes relating to lower spatial frequency attributes appear earlier

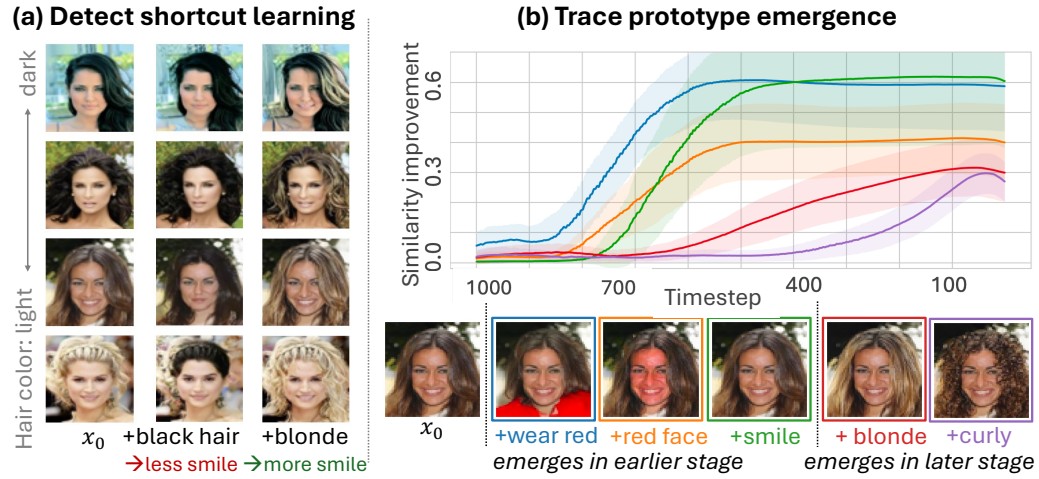

Figure 4: *Patronus* **as an interpretable tool** for (a) **Detecting shortcut learning.** Enhancing hair-color prototypes reveals their correlation with other attributes (e.g., smile), thereby exposing unwanted biases. (b) **Tracing prototype emergence.** Different prototypes appear at different stages of the diffusion process, as indicated by similarity score improvements, which provides insights for more effective image editing strategies.

during generation, such as "wearing red"; while higher spatial frequency attributes, like "curly hair", emerge later in the generation process. This insight could potentially improve the efficiency of image editing or counterfactual generation by guiding how far an image should be reversed in the diffusion process. It could also support bias mitigation by leveraging the same mechanism once unwanted correlations are detected.

**Prototype Correlation and Collapse.** We test whether correlation between prototypes is caused by *prototype collapse*, where multiple prototypes represent the same semantics. To assess this, we introduce a Prototype Distinct Loss to encourage prototype disentanglement and evaluate its impact compared to using the denoiser loss alone. The Prototype Distinct Loss is defined as: $\mathcal{L}_{distinct} = \frac{1}{N} \sum_{i=1}^{N} \max(0, \delta - \min_{j \neq i} D_{ij})$, where $D$ is the cosine distance with absolute similarity: $D_{ij} = 1 - |\frac{p_i \cdot p_j}{\|p_i\| \|p_j\|}|$. The margin $\delta$ is set to 0.5 and 1.0, where $\delta = 1.0$ enforces prototypes to be orthogonal. We initialize the new model using the original network parameters and train it for an additional 100 epochs using $L_{ddpm} + L_{distinct}$. Results show that the new models do not see a substantial change in the learned prototypes, suggesting that the prototypes optimized via the denoising objective are already sufficiently decorrelated without explicit regularization (see Appendix).

**Do We Capture All Relevant Attributes?** While *Patronus* shows great ability in capturing attributes, we notice that global features, e.g. gender and age, are harder to find in one specific prototype. This could result from the patch-based prototypical encoder, making non-local features hard to capture. See Appendix for illustrative visual examples.

## 5.1 CONCLUSION

We propose *Patronus*, an interpretable diffusion model integrated with a prototypical network. It enables intuitive interpretation of the generation process by visualizing learned prototypes (*what*) and identifying *where* and *when* they appear. It also supports semantic manipulation through prototype activation vector. Experiments show that *Patronus* achieves competitive performance and learns meaningful prototype-based representations. We further explore its capability to diagnose unwanted correlations in the generative process. We believe *Patronus* offers valuable insights into interpretable diffusion models by bridging diffusion and prototypical networks.

ACKNOWLEDGEMENTS.

Work on this project was partially funded by DTU Compute, the Technical University of Denmark; the Pioneer Centre for AI (DNRF grant nr P1); and the Novo Nordisk Foundation through the Center for Basic Machine Learning Research in Life Science (MLLS, grant NNF20OC0062606). The funding agencies had no influence on the writing of the manuscript nor on the decision to submit it for publication.

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

# *Patronus*: INTERPRETABLE DIFFUSION MODELS WITH PROTOTYPES

Nina Weng, Aasa Feragen, Siavash Bigdeli
Technical University of Denmark
`{ninwe, afhar, sarbi}@dtu.dk`

Code is available at nina-weng.github.io/patronus.github.io.

## APPENDIX

# A    ADDITIONAL IMPLEMENTATION DETAILS

## A.1    INTERPRETABILITY: LOCATING MOST ACTIVATED PATCHES

Given an input image $x_0$ and a chosen prototype $J$, the most activated patch is determined as follows:

1. Compute the similarity score between each latent feature $z_i$, where $i \in \{1, 2, \ldots, N\}$, and the prototype $p_J$. Identify the closest $z_i$ as:

$$i' = \arg \min_i D(z_i, p_J)$$

where $D(\cdot, \cdot)$ represents the chosen distance metric; in our case, it's the log transformed square $L2$ distance (Sec. 3.1).

2. Map the index $i'$ back to the spatial coordinates of the feature map. Using the receptive field of the encoder, locate the corresponding patch in the original image, as shown in Fig. 5.

## A.2    INTERPRETABILITY: EMERGENCE ANALYSIS

**Retrieve prototype activation vector along generation process for one image.**    For a given guidance $s$ and starting noise $x_T$, we can sample a generated image $x_0$, where $x_t$ denotes the intermediate state at each timestep. At each timestep $t$, the estimated denoised image $\hat{x}_0^t$ is computed as:

$$\hat{x}_0^t = \frac{1}{\sqrt{\bar{\alpha}_t}}(x_t - \sqrt{1 - \bar{\alpha}_t} \cdot \epsilon_\theta(x_t, t, s)). \tag{8}$$

Here, the superscript $t$ indicates that this estimate originates from timestep $t$. The prototype activation vector at timestep $t$ is then obtained as: $s^t = Enc(\hat{x}_{0,t})$. By repeating this process across all timesteps $t$, we obtain the sequence $\{s^t\}_{t=0}^T$. Extracting the $j$-th index rom each activation vector yields $\{s_j^t\}_{t=0}^T$, which tracks how the $j$-th prototype is activated throughout the generation process for sample $x_0$.

**Examples of how prototypes emerge differently over time during diffusion.**    Building on the illustrative example in Fig.1a, where specific semantic features are enhanced or suppressed using prototype activation vectors, we examine how the five selected prototypes emerge over time during the generation process. This is shown for the original image (Fig.6a), with the enhancement of the prototype "White collar" (Fig.6b), and with the enhancement of the prototype "Curly hair" (Fig.6c). Interestingly, when the "White collar" prototype is enhanced, its similarity score increases notably around timestep 700/1000 (Fig.6b). In contrast, when enhancing the "Curly hair" prototype, its similarity score begins to rise around timestep 200/1000 (Fig.6c). Note that larger $t$ corresponds to an earlier stage of the diffusion process, which means "Curly hair" as a semantic information emerged later in diffusion generation process, compared to "White collar".

This observation is further confirmed by the estimated denoised $\hat{x}_0^t$ in Fig.6b and Fig.6c. White clothing appears at an early stage of the denoised images, whereas although the hair becomes fluffier, the fine-grained curly texture only emerges around timestep 200 (bottom of Fig.6c). This difference

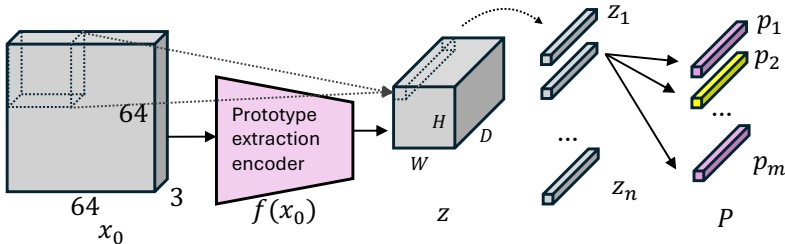

Figure 5: **Illustration on how to find the most activated patch given $x_0$ regarding prototype $J$.** Here the shape of $x_0$ ($64 \times 64 \times 3$) serves as an example and should be generalizable to other scenarios.

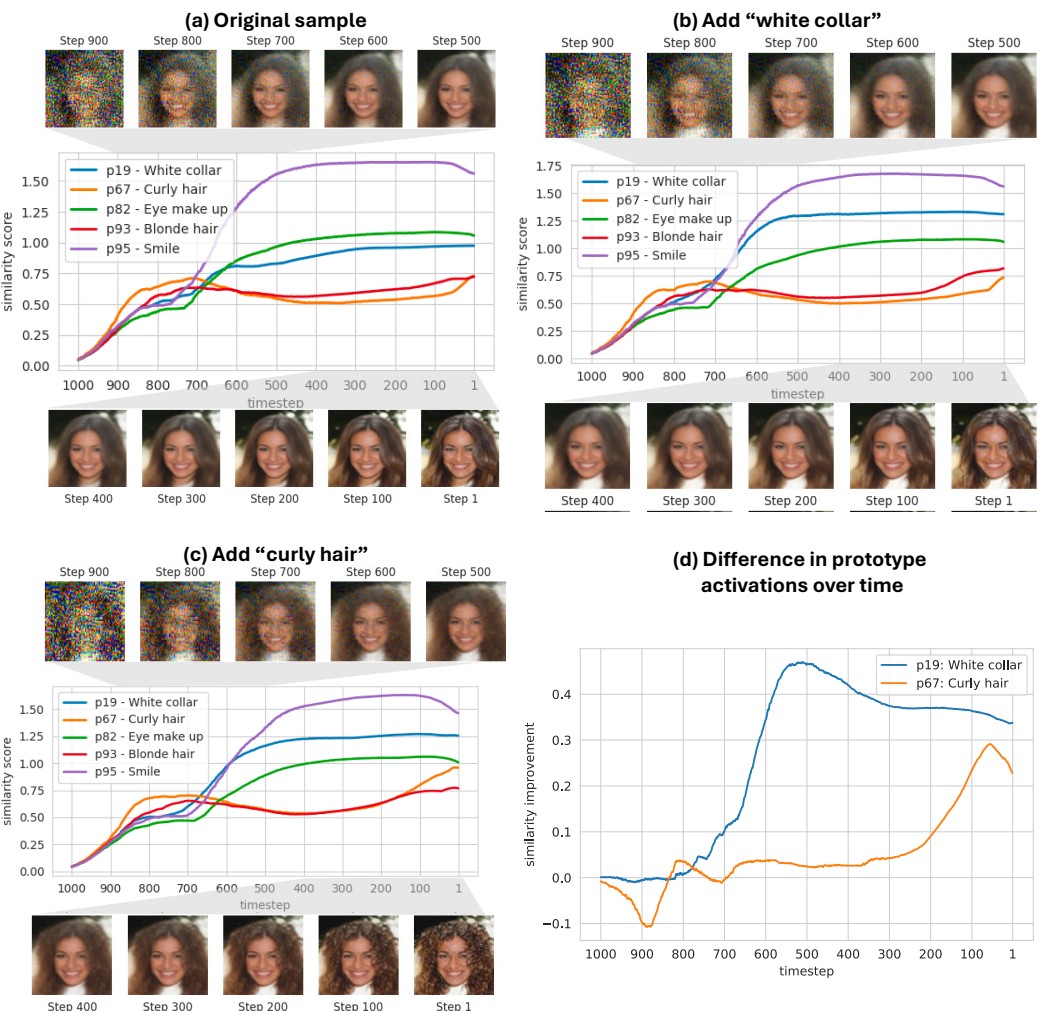

Figure 6: **Prototype emergence details in:** (a) **the original generated sample;** (b) generated sample with **"white collar"** added; (c) generated sample with **"curly hair"** added; (d) **differences in prototype activations over time**. Sharp increases indicate the timesteps when prototypes begin to emerge prominently in the enhanced sample compared to the original.

is summarized in Fig.6d, where we show the difference in prototype activation vectors between the enhanced and original samples. The sharp increases in the plotted curves indicate the timesteps at which the respective prototypes begin to emerge prominently.

**Trend analysis on a larger sample set.** To analyze the trend more comprehensively, we randomly select 100 samples from the test set and enhance all 100 prototypes for each sample. We then compute the average difference between the enhanced activation scores and their original values, visualizing the results as an averaged curve, just as shown in Fig. 4b.

### A.3 HYPER-PARAMETERS FOR TRAINING

More hyper-parameter for Patronus training could be found in Table 3. We trained all experiments with Adam optimizer, learning rate of $1e^{-4}$.

For the latent diffusion model, we trained with an 1D UNet, with base channels of 64, and channel multipliers as (1,2,4). We set the dropout rate as 0.2 and learning rate as $1e^{-4}$.

Table 3: **Hyperparameters used for *Patronus* training.**

|  | Input size | Num. p | shape of p | patch size | Num. patches | Num. Channels | Num. Channel. Mult |
|---|---|---|---|---|---|---|---|
| FashionMNIST | (1,32,32) | 30 | (1,1,128) | 14×14 | 100 | 64 | 1,2,4,4 |
| Cifar10 | (3,32,32) | 100 | (1,1,128) | 14×14 | 100 | 64 | 1,2,4,4 |
| FFHQ | (3,64,64) | 100 | (1,1,128) | 14×14 | 672 | 64 | 1,2,4,4 |
| CelebA | (3,64,64) | 100 | (1,1,128) | 14×14 | 672 | 64 | 1,2,4,4 |
| CheXpert | (1,224,224) | 100 | (1,1,128) | 60×60 | 1849 | 64 | 1,2,4,4 |

Table 4: **Ablation study on CelebA**, regarding the number of prototypes and the prototype vector size.

| # p | shape of p | TAD ↑ | Attrs ↑ | Latent AUROC↑ | FID ↓ |
|---|---|---|---|---|---|
| 32 |  | $0.8291 \pm 0.0146$ | $9.0000 \pm 0.0000$ | $0.8288 \pm 0.0022$ | $6.0126 \pm 0.0740$ |
| 64 | (1,1,128) | $\mathbf{0.8515 \pm 0.0545}$ | $9.2000 \pm 0.4000$ | $0.8527 \pm 0.0017$ | $6.5395 \pm 0.0676$ |
| 100 |  | $0.5395 \pm 0.1205$ | $12.0000 \pm 1.0954$ | $0.8646 \pm 0.0010$ | $5.4871 \pm 0.0151$ |
| 128 |  | $0.4700 \pm 0.0758$ | $\mathbf{12.0000 \pm 0.8944}$ | $0.8713 \pm 0.0018$ | $\mathbf{5.1264 \pm 0.0488}$ |
| | (1,1,64) | $0.5860 \pm 0.0535$ | $8.4000 \pm 0.4899$ | $0.8491 \pm 0.0021$ | $5.2017 \pm 0.0429$ |
| 64 | (1,1,128) * | $\mathbf{0.8515 \pm 0.0545}$ | $9.2000 \pm 0.4000$ | $0.8527 \pm 0.0017$ | $6.5395 \pm 0.0676$ |
| | (1,1,256) | $0.4779 \pm 0.0045$ | $8.0000 \pm 0.0000$ | $0.8492 \pm 0.0008$ | $6.2918 \pm 0.1279$ |

\* The result is the same as the second row since the hyper-parameters are identical. We listed it as another row for easier comparison.

For prototype encoder, in this work we apply a 4-layer convolutional network with ReLU activations. The channel progression is $1 \rightarrow 32 \rightarrow 64 \rightarrow 64 \rightarrow 128$. All convolutional layers use a $3 \times 3$ kernel, with strides of $[2, 1, 1, 1]$ for the four layers and paddings of $[1, 0, 0, 0]$, respectively.

### A.4 CheXpert Dataset

In this work, we use a subset of the CheXpert dataset (Irvin et al., 2019), retaining only frontal chest X-ray scans. To mitigate potential information leakage and reduce memorization effects due to patient-specific variations, we sample a single scan per patient[2]. This preprocessing step yields a total of 28,878 chest X-rays, of which 90% are allocated for training and the remaining 10% for testing.

## B Ablation Study

As shown in Tab. 4, we present an ablation study of *Patronus* with respect to the number of prototypes and the dimensionality of the prototype vectors. Experiments are conducted on the CelebA dataset using an input resolution of $(3, 64, 64)$, a training duration of 200 epochs, and a learning rate of $1e-4$. Note that FID is computed in the context of conditional generation.

### B.1 Number of Prototypes

We evaluate the impact of varying the number of prototypes, setting $\#p = \{32, 64, 100, 128\}$, while fixing the prototype vector size to $(1, 1, 128)$. As the number of prototypes increases, we observe consistent improvements in latent quality (measured by AUROC), the number of attributes captured, and the FID score. However, the TAD score—which reflects the disentanglement quality—tends to decline. This trade-off is expected: while a larger prototype pool allows the model to capture more fine-grained visual patterns, it also introduces redundancy, reducing the distinctiveness and interpretability of individual prototypes. This suggests the existence of an optimal prototype budget for balancing generation quality and disentanglement.

---

[2]The number of recordings per patient in CheXpert is highly imbalanced, ranging from 1 to 89 (Weng et al., 2023). Notably, disease severity is correlated with scan frequency—fewer than 25% of control subjects have more than five scans, while this proportion exceeds 50% among patients.

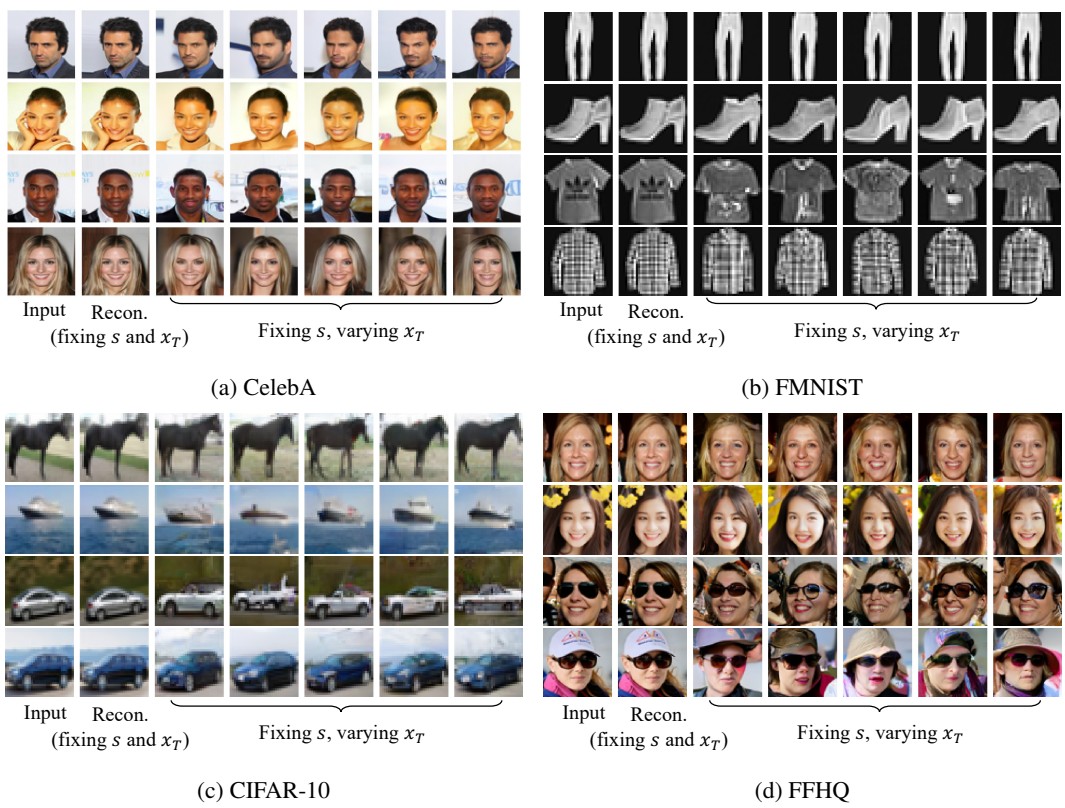

(a) CelebA

(b) FMNIST

(c) CIFAR-10

(d) FFHQ

Figure 7: **Additional examples of reconstruction and variations with fixed $s$ and random $x_T$.**

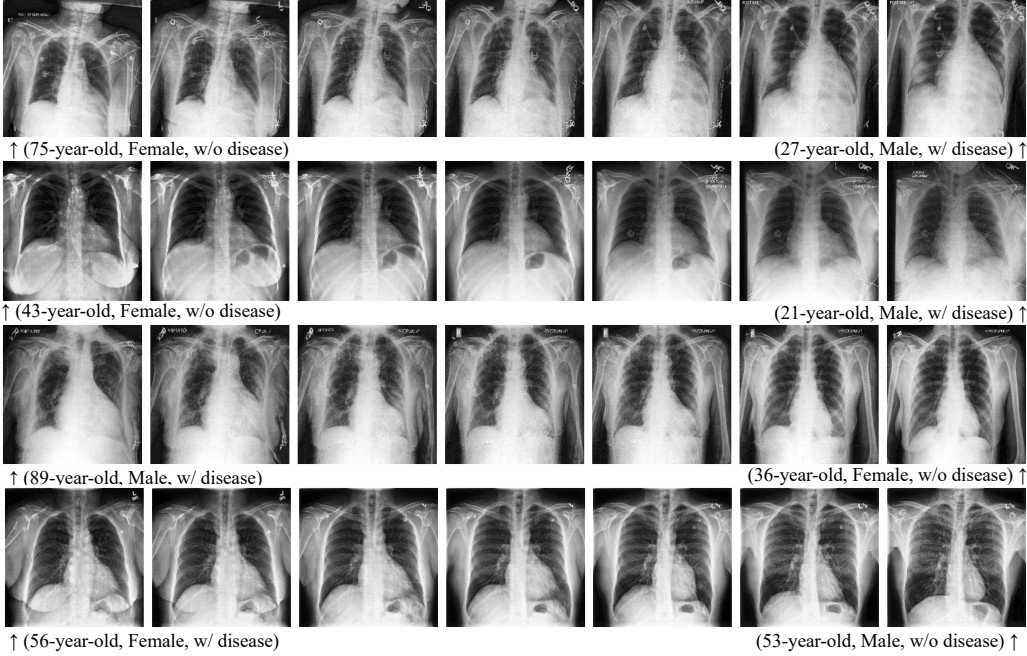

↑ (75-year-old, Female, w/o disease)     (27-year-old, Male, w/ disease) ↑

↑ (43-year-old, Female, w/o disease)     (21-year-old, Male, w/ disease) ↑

↑ (89-year-old, Male, w/ disease)     (36-year-old, Female, w/o disease) ↑

↑ (56-year-old, Female, w/ disease)     (53-year-old, Male, w/o disease) ↑

Figure 8: **Additional Image Interpolation Examples on CheXpert.** Interpolations are shown between two real chest X-rays, with patient age, gender, and disease condition (Cardiomegaly).

## B.2 Prototype Vector Size

A larger prototype vector size allows each prototype to encode more detailed semantic information within a fixed spatial region. Conversely, when the prototype vector size is too small, the model may struggle to capture sufficient semantic richness. However, excessively large prototype vectors may introduce optimization challenges, such as slower convergence and increased redundancy. This trade-off is reflected in the results shown in Tab. 4, where a prototype vector size of $(1, 1, 128)$ yields the best overall performance.

## C Additional Experimental Results

### C.1 Additional Visual Examples

**Reconstruction and Variation with Random** $x_T$    We provide additional visual examples across multiple datasets in Fig. 7. These results demonstrate *Patronus*' ability to capture semantic features across different datasets.

Notably, while *Patronus* effectively preserves fine details and patterns, it struggles with rare patterns. For instance, in the third row of Fig.7b, the model fails to reconstruct the Adidas logo accurately. Another interesting case appears in the last row of Fig.7d, where *Patronus* successfully identifies the presence of a hat but generates variations of different hat styles instead of an exact reproduction.

**Interpolation**    We provide additional visual examples of interpolation across various datasets, including CheXpert (Fig.8), as well as FMNIST, CIFAR-10, CelebA, and FFHQ (Fig.9a). In datasets with well-defined class clusters, such as Fashion-MNIST and CIFAR-10, interpolation tends to be less effective when transitioning between images belonging to different classes (Fig. 9b).

**More Visual Samples for Diagnosis Ability of Patronus**    In Fig.10, we present additional samples and hair-color-related prototypes for the diagnosis task, revealing a more pronounced bias—enhancing hair color also affects the presence of a smile. More specifically, we showcase eight cases, including four female and four male subjects. Within each gender group, we include two individuals with black hair and no smile, alongside one individual with brown or blonde hair and a smile. When enhancing black hair-related prototypes, all images transition to a non-smiling expression (as seen in the fourth and sixth columns of Fig.10).

### C.2 Visual Representation of Prototypes

We present a complete visual representation of the learned prototypes from the CelebA dataset in Fig. 11 and 12. Below each prototype, we provide a summary of its semantic meaning based on human observation without explicit annotation. Consequently, these interpretations may contain inaccuracies. For prototypes where a clear semantic meaning could not be determined, we leave the description blank. Notably, these blank descriptions highlight the inherent limitations of language in capturing visual concepts. The visualization process follows the steps outlined in Sec. 3.3.

### C.3 Captured Attributes by a Single Prototype

As shown in Tab. 1, we applied TAD and the number of attributes captured to estimate the prototype disentanglement ability, where *Patronus* remarkably outperformed the SOTA by nine captured attributes, while prior methods capture at most three. Tab. 5 details which prototypes capture these attributes, with visualizations in Fig. 13.

As shown in Tab. 5, a single prototype can capture multiple attributes. E.g. prototype 82 captures both "Eyeglasses" and "Rosy_Cheeks". This finding is particularly interesting, as prior visual inspection suggested that prototype 82 represents the concept of "Heavy Eye Make-up" (Fig. 3), which is semantically related to both attributes: heavy eye make-up often co-occurs with rosy cheeks and tends to be negatively correlated with eyeglasses, possibly because individuals wearing heavy makeup are more likely to use contact lenses instead. This observation is further supported by the intervention results shown in Fig. 13, where suppressing the activation of $p_{82}$ leads to the appearance of eyeglasses, while enhancing the activation induces both rosy cheeks and prominent eye make-up.

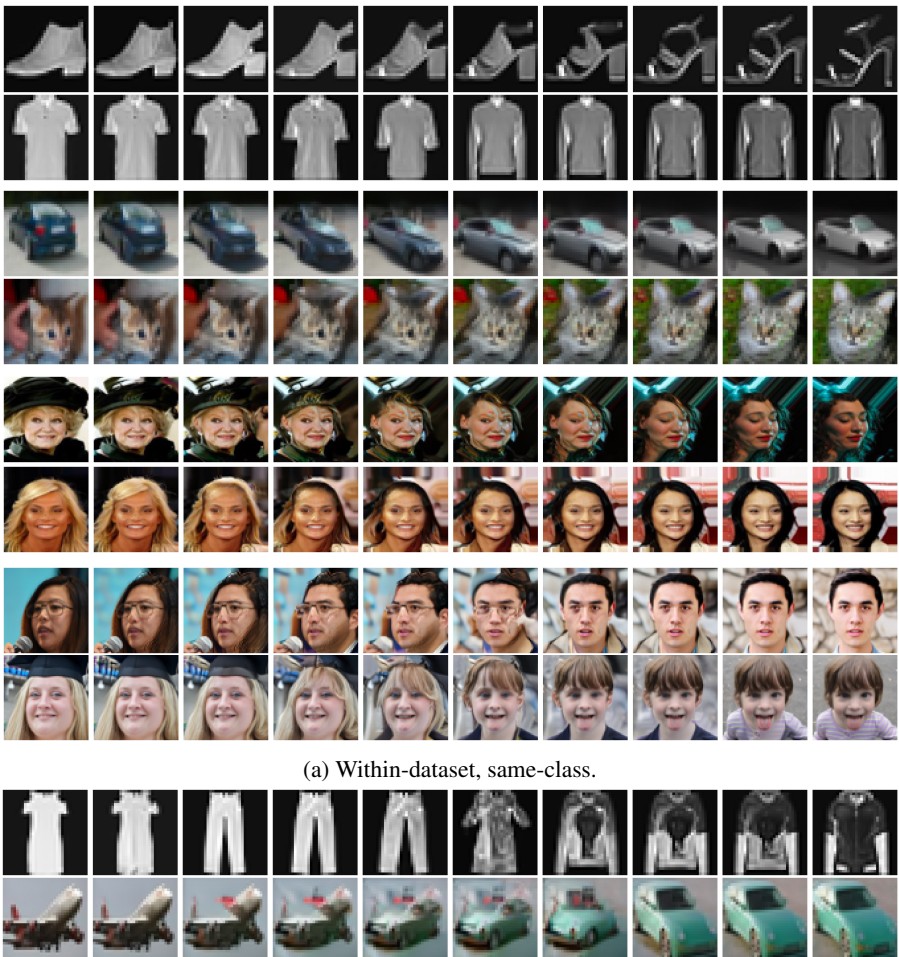

(a) Within-dataset, same-class.

(b) Across classes.

Figure 9: **Additional interpolation examples.** (a) Interpolation across datasets: Fashion-MNIST (first two rows), CIFAR-10 (third and fourth), CelebA (fifth and sixth), FFHQ (seventh and eighth) (b) Cross-class interpolation (Fashion-MNIST, CIFAR-10) showing incoherent transitions.

Table 5: **Captured Attributes in CelebA.**

| Captured Attributes | Captured Attributes AUROC | Captured Prototype Index |
|---|---|---|
| Bald | $0.8276 \pm 0.0041$ | 30 |
| Bangs | $0.8411 \pm 0.0022$ | 9 |
| Black_Hair | $0.8104 \pm 0.0034$ | 78 |
| Blond_Hair | $0.8917 \pm 0.0022$ | 93 |
| Blurry | $0.8717 \pm 0.0058$ | 35 |
| Eyeglasses | $0.7967 \pm 0.0026$ | 82 |
| Pale_Skin | $0.8549 \pm 0.0044$ | 58 |
| Rosy_Cheeks | $0.8226 \pm 0.0040$ | 82 |
| Wearing_Hat | $0.9002 \pm 0.0025$ | 9 |

## C.4 Latent Quality for CheXpert Dataset

We analyze the latent quality for CheXpert dataset by measuring TAD, number of attributes being captured and latent AUROC in Tab. 6. A total of 23 attributes are evaluated, comprising four demographic attributes, four indicators related to patients' socioeconomic or health status, one shortcut

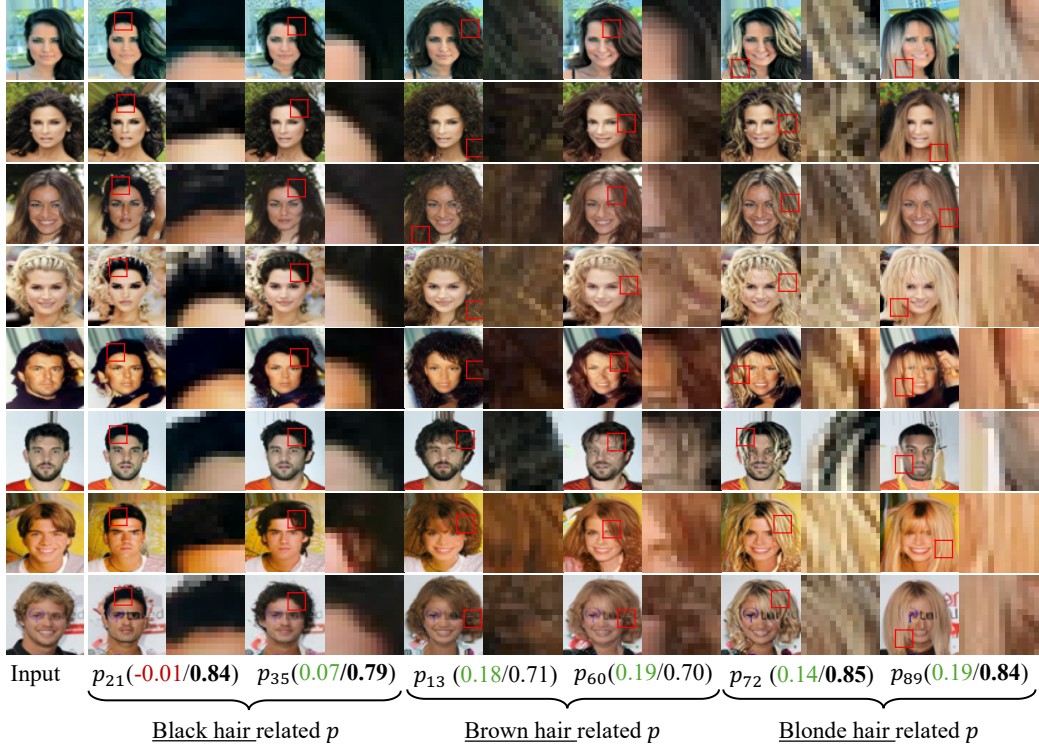

Input    $p_{21}$(-0.01/**0.84**)   $p_{35}$(0.07/**0.79**)    $p_{13}$ (0.18/0.71)   $p_{60}$(0.19/0.70)    $p_{72}$ (0.14/**0.85**)   $p_{89}$(0.19/**0.84**)

Black hair related $p$     Brown hair related $p$     Blonde hair related $p$

Figure 10: **More visual samples and more hair-color related prototypes for diagnosing unwanted correlations with *Patronus***. By selecting the top two hair color prototypes (highest AUROC for single-dimension prediction), their correlation with "smile" prototype reveals dataset bias. $p_j(a/b)$ define as: (a) Spearman correlation with the "smile" prototype (green: positive, red: negative), and (b) AUROC for predicting hair color when using this prototype (**bold** if ≥ 0.75).

feature pacemaker (annotated by Weng et al. (2024)), and 14 disease-related labels. All attributes are binarized as detailed below[3]: Age (≥60 or <60), Sex (Male or Female), Race (White or Non-white), Ethnicity (Hispanic/Latino or Other), Insurance (Enrolled in Medicare or Not), Interpreter Need (Yes or No), Deceased (Yes or No), and BMI (within the normal range: 18.5–25.0, or outside).

**Evaluating latent quality.** As shown in Tab. 6, the learned latent representations demonstrate strong predictive capabilities for most demographic attributes, such as age, sex and BMI, achieving high AUROC scores using a simple logistic regression model. Notably, the latent space also encodes information relevant to the presence of a pacemaker. Furthermore, it supports the prediction of several cardiopulmonary conditions—such as Cardiomegaly, Edema, and Pleural Effusion—with AUROC values exceeding 0.75 (bold font in the table). These results indicate that the latent representations capture semantically meaningful and clinically relevant information. For comparison, we include the performance of a ResNet-50 baseline in the Tab. 6. It is worth noting that this baseline was trained on 320×320 resolution images (Bressem et al., 2020), and that the original CheXpert validation set includes only samples from five disease categories. As a result, performance metrics for the remaining categories are unavailable.

**Interpreting Captured Attributes and Their Corresponding Prototypes.** Among the 23 attributes, three are captured by a single prototype, as listed in Tab. 6, with visualizations provided in Fig. 14. Notably, the **Sex** attribute is captured by a prototype that focuses on the edge of the chest wall (Fig. 14a). In the extrapolation experiment on prototype $p_1$ (Fig. 14b), we observe a decrease in rib cage size as the prototype similarity score increases. This observation aligns with clinical find-

---

[3]We acknowledge that the binarization is not ideal, as it may be white-centralized and introduce bias; however, it is adopted here for the sake of simplification.

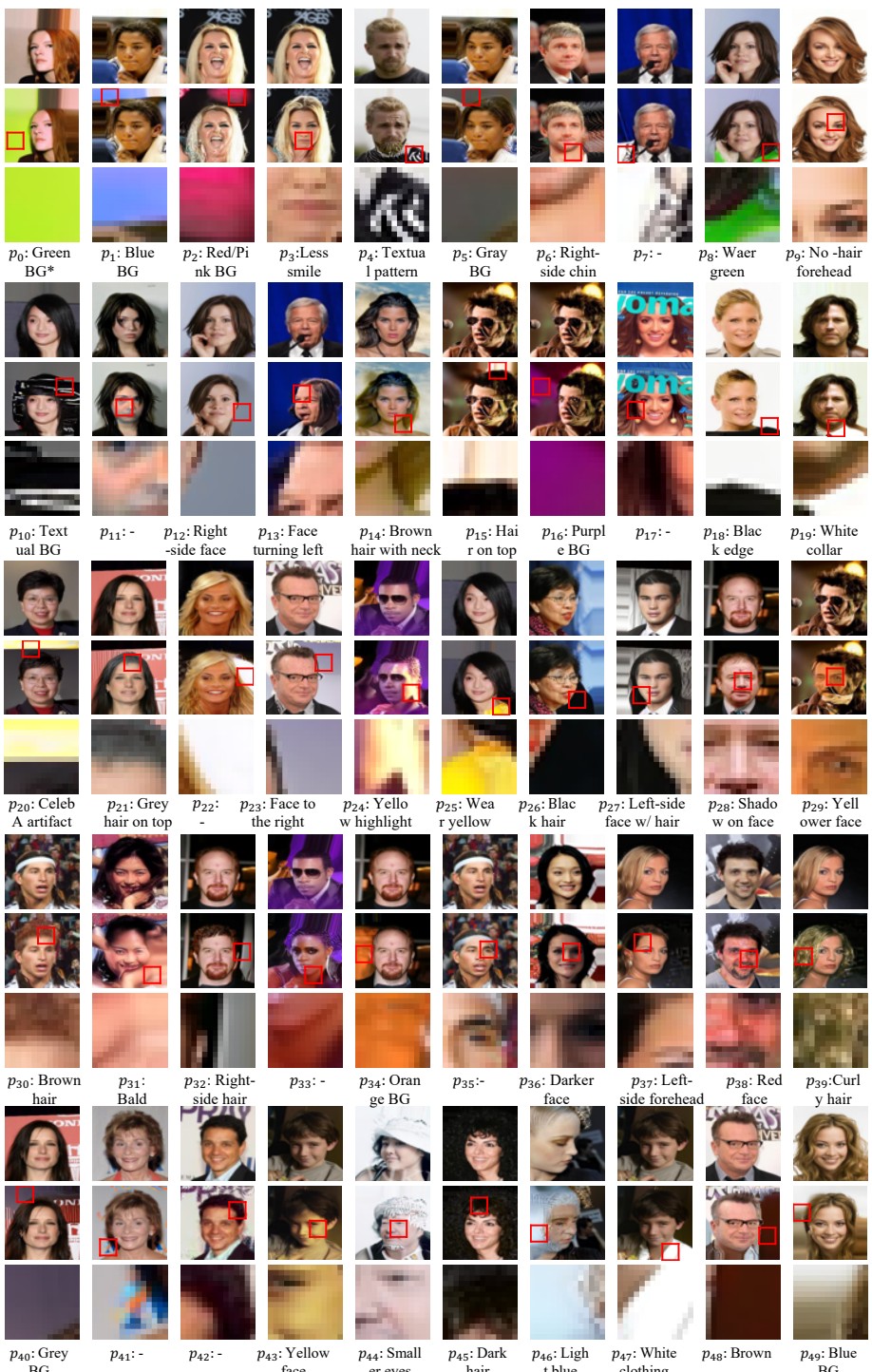

Figure 11: **Visualization of learned prototypes on CelebA (prototypes 0 - 49).** Each row shows 10 prototypes with three views per prototype (top to bottom): original image, image enhanced with prototype $j$, and most activated patch (serves as the prototype visualization). *BG* means background.

ings that females tend to have a disproportionately smaller rib cage compared to males (Bellemare et al., 2003; 2001). For the **Age** attribute, the corresponding prototype is most activated in the region around the upper thoracic vertebrae (Fig. 14a). This may relate to the age-associated ossification

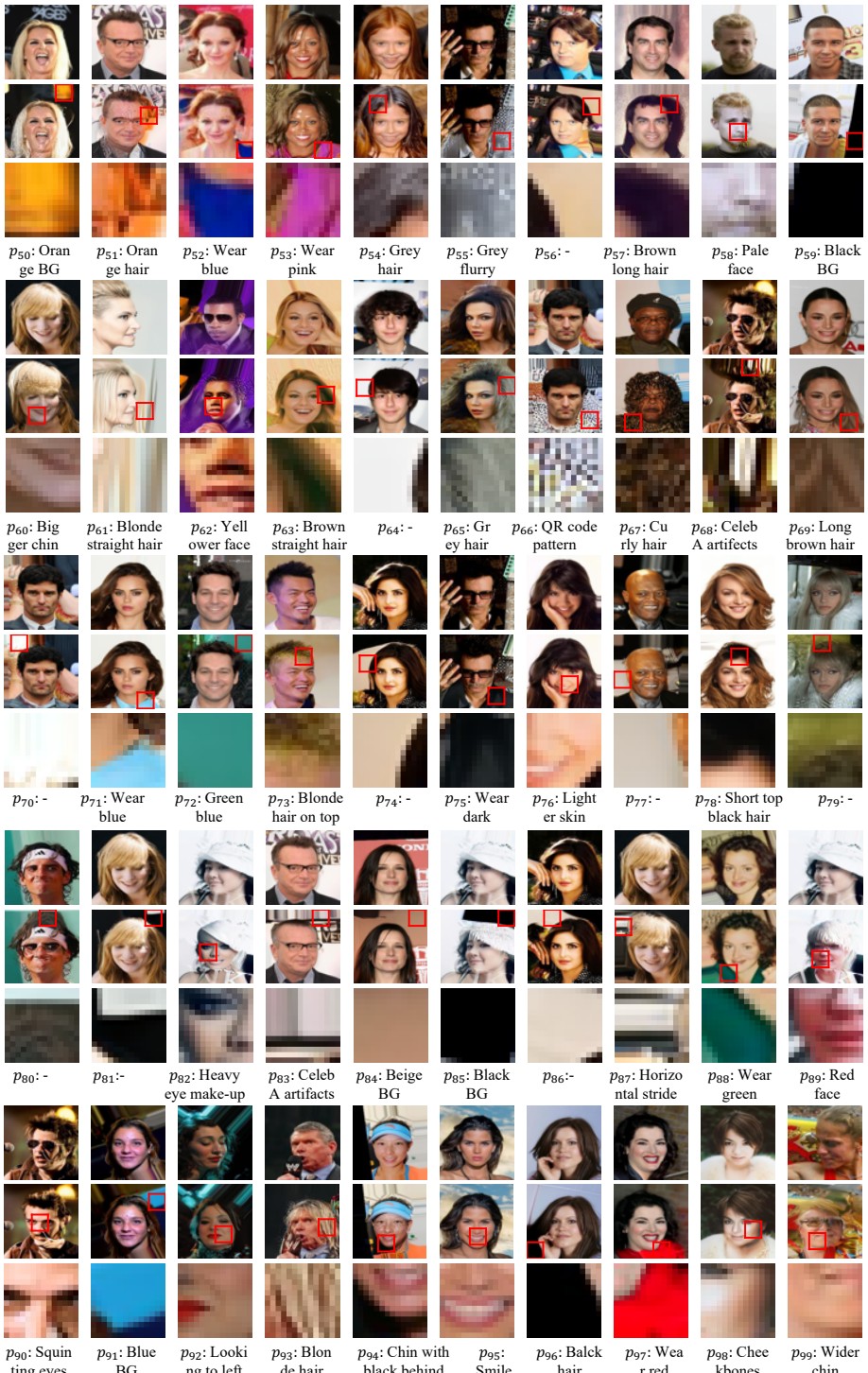

Figure 12: **Visualization of learned prototypes on CelebA (prototypes 50 - 99).** Each row shows 10 prototypes with three views per prototype (top to bottom): original image, image enhanced with prototype $j$, and most activated patch (serves as the prototype visualization). BG means background.

of the costochondral cartilage of the first rib (McCormick & Stewart, 1988; Radiology Masterclass, n.d.). Regarding the **Pacemaker** attribute, the most activated patch is located near the upper region of the heart, potentially reflecting the correlation between pacemaker presence and underlying car-

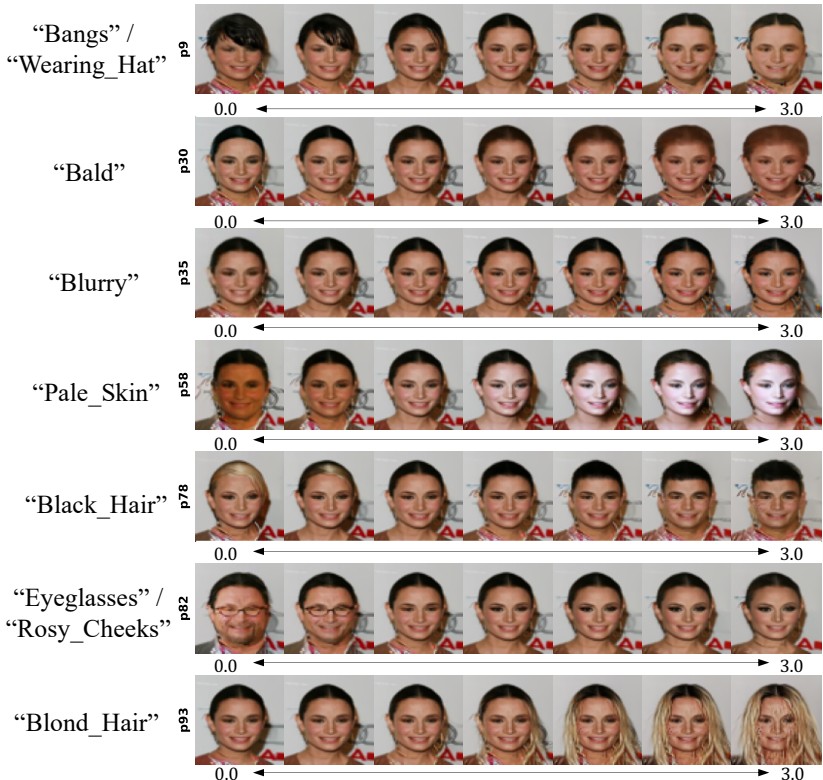

Figure 13: **Visualization of Captured Attributes in CelebA via Explorations.** The text on the left indicates the attributes captured in the CelebA dataset. Note that a single prototype can potentially capture multiple attributes.

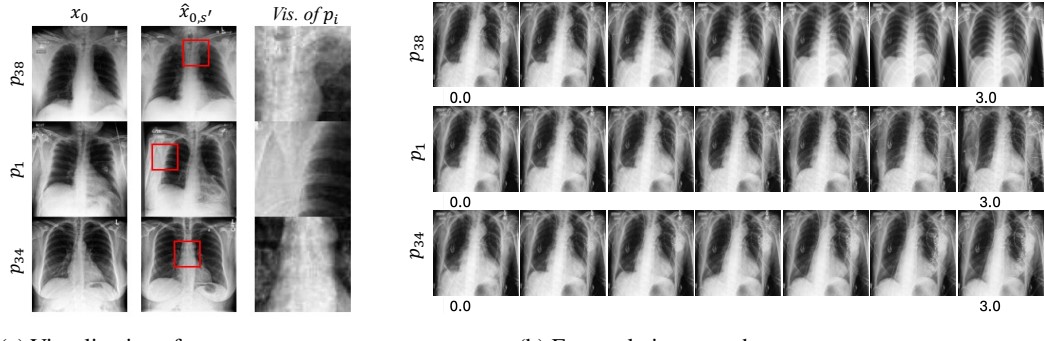

(a) Visualization of prototypes capturing attributes.

(b) Extrapolation over the same prototypes.

Figure 14: **Visualization and extrapolation of prototypes capturing attributes from CheXpert.** Three attributes are captured: Age (by $p_{38}$), Sex (by $p_1$), and presence of a pacemaker (by $p_{34}$).

diac conditions. In Fig. 14b, increasing the activation of prototype $p_{34}$ leads to a more pronounced appearance of a pacemaker.

We acknowledge the limitations of our medical expertise and do not intend to draw definitive clinical conclusions from these observations. We welcome researchers with medical backgrounds to further evaluate and interpret these findings.

Table 6: **Latent Quality of the CheXpert Dataset.** The *latent AUROC* denotes the average AUROC across all considered attributes. A total of 23 attributes are evaluated, comprising four demographic attributes, four indicators related to patients' socioeconomic or health status, one shortcut feature (pacemaker), and 14 disease-related labels. All attributes are binarized. **Bolded** values indicate AUROC $\geq 0.75$.

| TAD↑ | Attrs ↑ | Latent AUROC↑ | Demographic Attributes* | | | | Other Attributes* | | | | Shortcut PM [†] |
|---|---|---|---|---|---|---|---|---|---|---|---|
| | | | Age | Sex | Race | Ethnicity | Insurance | Interpreter | Deceased | BMI | |
| 0.12±0.01 | 3.0±0.0 | 0.74±0.01 | **0.88±0.01** | **0.98±0.00** | 0.70±0.01 | 0.72±0.01 | 0.74±0.00 | 0.74±0.00 | 0.66±0.01 | **0.77±0.01** | **0.92±0.01** |

| | No Finding | Enlarged CM[‡] | Cardio-megaly | Lung Opacity | Lung Lesion | Edema | Consoli-dation | Pneu-monia | Atelec-tasis | Pneumo-thorax | Pleural Effusion | Pleural Other | Fracture | Support Devices |
|---|---|---|---|---|---|---|---|---|---|---|---|---|---|---|
| | | | | | | | **Disease Labels** | | | | | | | |
| LR using $p$ | **0.86** | 0.61 | **0.75** | 0.70 | 0.65 | **0.77** | 0.64 | 0.61 | 0.60 | 0.68 | **0.80** | 0.70 | 0.68 | **0.76** |
| ResNet50 | - | - | 0.80 | - | - | 0.88 | 0.90 | - | 0.80 | - | 0.91 | - | - | - |

| Captured Attributes | Captured Attributes AUROC | Captured Prototype Index |
|---|---|---|
| Age Group | $0.7891 \pm 0.0031$ | 38 |
| Sex | $0.8806 \pm 0.0005$ | 1 |
| PM[†] | $0.7823 \pm 0.0028$ | 34 |

* AUROC Performance from Latent Representations. The reported AUROC values reflect the performance of a logistic regression classifier trained on the latent representations using 5-fold cross-validation. Demographic and other attributes are binarized, details are in text.
[†] Pacemaker, annotations from Weng et al. (2024).
[‡] Enlarged Cardiomediastinum.

## C.5    PROTOTYPE CONSISTENCY

**Prototype visualization consistency for one run.**    We quantitatively evaluated the consistency of prototype visualizations in one run. As the visual concepts are very small patches, consistency is measured directly in pixel space, which faithfully captures their low-level structural differences.

Using Euclidean (L2) distance between visualization pairs, we observe a clear separation: within-prototype = 2.31 vs. between-prototype = 3.95. A two-sample t-test confirms this separation is highly significant (t = 19.55, p = $1.6 \times 10^{-37}$). This confirms that each prototype forms a coherent and distinguishable visual concept.

**Learned prototype consistency cross different run.**    This part evaluates the consistency of learned prototypes across different random initializations. We'd like to emphasis that our goal is not to enforce identical explanations across models, but to explain each model's behaviour as it is trained. If two models use different internal concepts during generation, then different prototypes are expected and even desirable. Nevertheless, under almost identical training conditions (same architecture, dataset, and objective), it is still meaningful to assess whether the learned prototypes remain consistent.

To do so, we compare prototype activation patterns on the same input batches rather than directly comparing prototype vectors, since the latter are not aligned across runs due to arbitrary rotations in the feature space. For each batch of $B$ images (here $B = 512$), we obtain prototype activations from both models:

$$A^1, A^2 \in \mathbb{R}^{N \times B},$$

where each row corresponds to the activation pattern of one prototype across the batch.

We then compute the pairwise cosine similarity matrix:

$$S_{i,j} = \frac{\langle A_i^1, A_j^2 \rangle}{\|A_i^1\| \cdot \|A_j^2\|}.$$

To resolve the permutation ambiguity between prototype indices, we determine the optimal one-to-one alignment using the Hungarian algorithm :

$$\pi = \arg \max_{\text{1-1 mapping}} \sum_i S_{i,\pi(i)}.$$

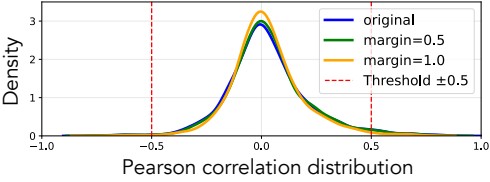

(a) **Distribution of Pearson correlation values for prototype similarity scores**, computed from 1,024 randomly selected training samples. Results compare the original training and training with an additional loss term. The difference is minimal, with only a small fraction of pairs exceeding an absolute correlation of 0.5 (red dashed lines) for all cases.

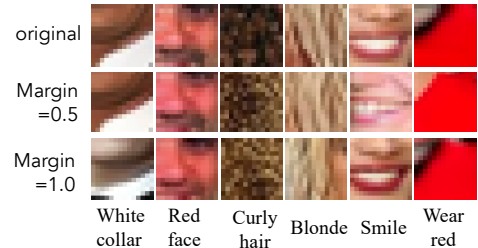

(b) Prototype visualizations and their semantic interpretations remain consistent regardless of the additional loss.

Figure 15: **Effect of additional loss on prototype diversity.** (Top) Pearson correlation distribution of prototype similarity scores shows minimal differences regardless of additional loss. (Bottom) Prototype visualizations remain consistent, indicating negligible impact on semantic representations.

We then assess how consistently this alignment appears across all test batches. The resulting permutation consistency score is 0.882, indicating that the prototype-to-prototype mapping across seeds is largely stable, with most prototypes consistently aligned between runs. After applying the optimal alignment, the matched prototypes achieve extremely high similarity (mean $\approx 0.99$), indicating that the semantic behaviour of prototypes is almost identical across seeds.

# D    EXTENDED DISCUSSION

## D.1    PROTOTYPE CORRELATION AND COLLAPSE

As discussed in Sec. 5, we further provide additional experimental results in Fig. 15. These results confirm that the new models do not show substantial changes in the learned prototypes, suggesting that prototypes optimized via the denoising objective are already sufficiently decorrelated without explicit regularization.

## D.2    LIMITATIONS OF CROSS-MODAL INTERPRETABILITY

We emphasize interpreting diffusion models *without* cross-modality by design for two key reasons:

**Language's limitation in representing complexity**: While language, as a discursive symbolism, serves as a powerful medium for interpretation, it alone cannot fully represent non-symbolic sensory and semantics complexity (Langer, 2009). This is evident in the superior performance of multimodal learning over single-modality; prior work (Gal et al., 2022; Goyal et al., 2017) further shows the restricted perceptual capacity of language-only generation by introducing visual cues.

**Inherited bias from the text embedding**: (1) Incomplete text representations: if certain concepts are not explicitly named (e.g., medical devices in radiology reports), the model cannot learn their visual counterparts. (2) Spurious correlations: text data may encode unintended biases, such as differing report detail levels by patient demographics, which may propagate into generated images.

## D.3    DO WE CAPTURE ALL RELEVANT ATTRIBUTES?

We further illustrate the challenge of capturing global features with sample results in Fig. 16. Following the experiment described in Sec. 4.1, we reconstruct images and their variations using a fixed $s$ with either fixed or random $x_T$. While fine-grained details, such as cheekbone structure in the second row and shirt details in the first row, are well preserved, the generated images with varying $x_T$ fail to capture age or gender consistently on harder scenarios. For instance, the middle image in the first row appears noticeably younger than the original, while most variations in the second row

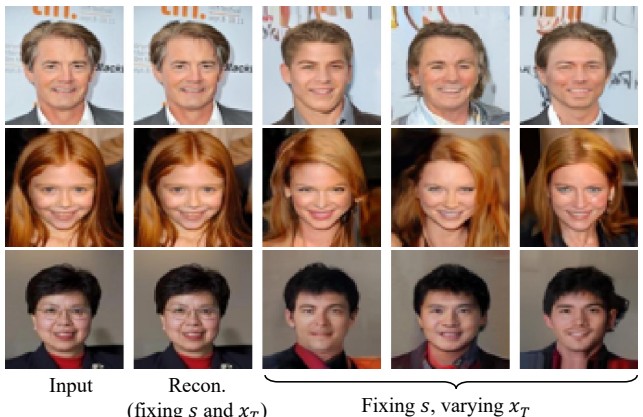

Input Recon. (fixing $s$ and $x_T$)  Fixing $s$, varying $x_T$

Figure 16: *Patronus* **may struggle to capture global features such as age and gender**. To illustrate this, we present three samples generated using the fixed $s$ with random $x_T$. While the generated images successfully preserve semantic attributes such as hair color, cheekbones, smile, shirt, and background color, they fail to accurately reconstruct age or gender.

depict a more mature appearance. Additionally, in the third sample, a gender shift from female to male is observed, possibly influenced by the presence of short hair.

### D.4 JUSTIFICATION FOR EXCLUDING PROTOPNET'S ADDITIONAL LOSS TERMS

In this work, we optimize the model using only the denoiser loss. A key question arises: *Is this loss sufficient?* Besides the attempt in Sec. 5 (Fig. 15) to add an extra disentanglement loss, we further compare with ProtoPNet, which, alongside cross-entropy loss for classification, introduces two additional loss terms: **(1) Cluster loss (Clst)**: encourages each training sample to have at least one patch close to a prototype; **(2) Separation loss (Sep)**: pushes latent patches away from prototypes of other classes.

Neither loss applies in our setting: (i) prototype representation is supposed to capture the underlying data distribution, *not* to enforce proximity to specific training samples; (ii) our method is generative and does not rely on class labels, making class-dependent separation constraints irrelevant. Thus, the denoiser loss is sufficient, as ProtoPNet's additional terms do not align with the objectives.

## E THE USE OF LLMS IN THIS WORK

In this work, we used large language models (LLMs) solely to assist with the presentation of the paper, including grammar correction, wording refinement, and minor sentence shortening (at the level of one or two words, not entire paragraphs). LLMs were not used for any other purpose. We always check the content after using LLMs, and we are responsible for the content that we submit.

## F QUANTITATIVE EVALUATION OVER INTERPRETABILITY

### F.1 EVALUATING PROTOTYPE–LANGUAGE ALIGNMENT VIA IMAGE CAPTIONING

An additional experiment was conducted to examine whether language-based analysis can support the alignment between visual concepts and human understanding. Specifically, we aim to assess whether our prototypes encode visual features that are both visible and interpretable through human concepts. To this end, we used the BLIP model (Salesforce/blip-image-captioning-large on Hugging-Face) to caption the prototype-enhanced image for prototype $j$ and the corresponding original image, and then compared the words with the most significant increases. For example, for the prototype corresponding to 'curly hair' in the visualization (Fig. 3a), the top three words with the highest increases after applying the prototype are 'curly': 409, 'hair': 307, 'long': 66, where each value

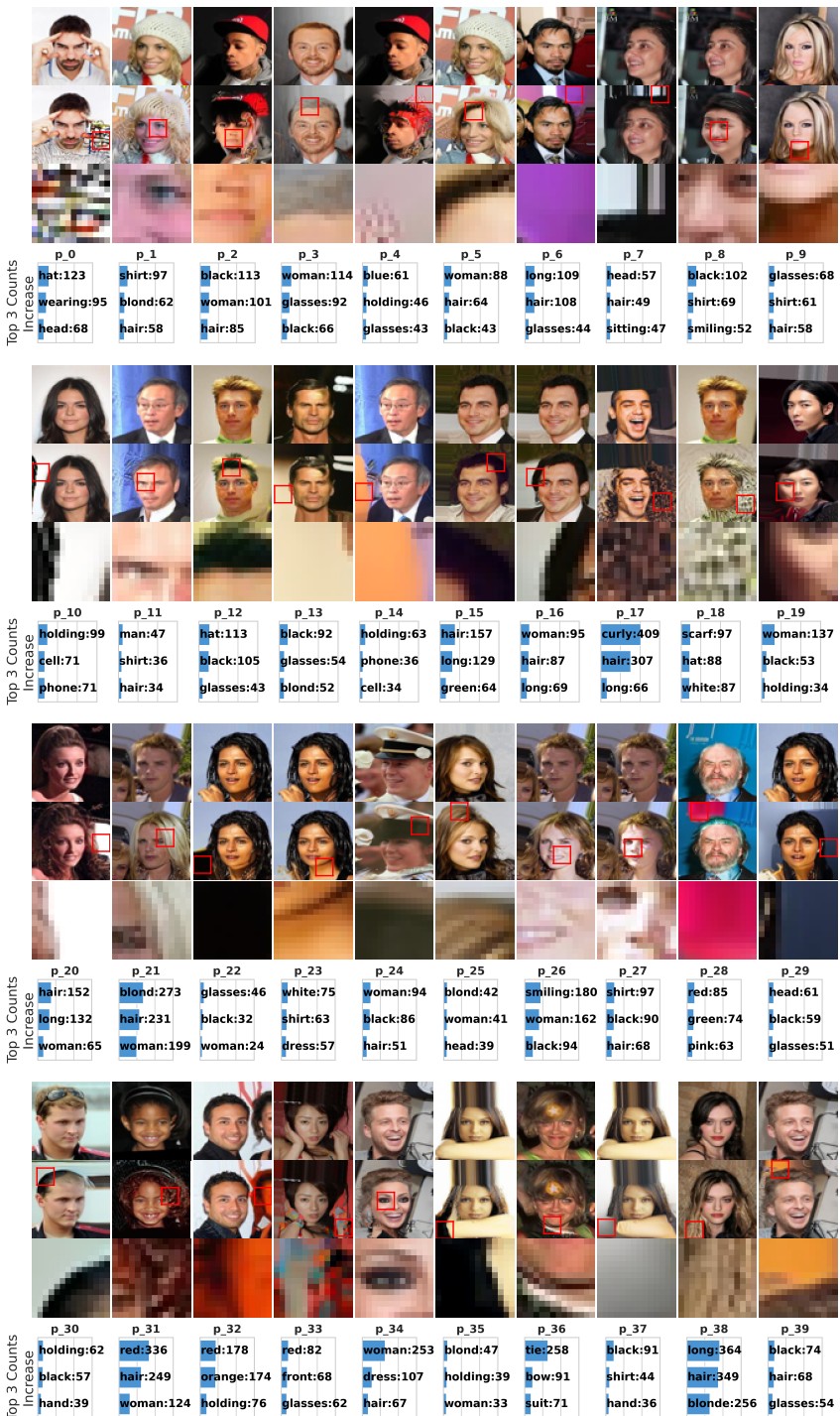

Figure 17: **Visualization of learned prototypes on CelebA with the top three increased words using BLIP for captioning (p0-p39).** Each row shows 10 prototypes with three views per prototype (top to bottom): original image, image enhanced with prototype $j$, and most activated patch (serves as the prototype visualization), and the top three increased words.

reflects the difference in word frequency between the enhanced and original images across 672 samples. We provide the full list of the top three increased words, along with prototype visualizations from one run, in Fig. 17-18. Note that this evaluation depends strongly on the captioning model's

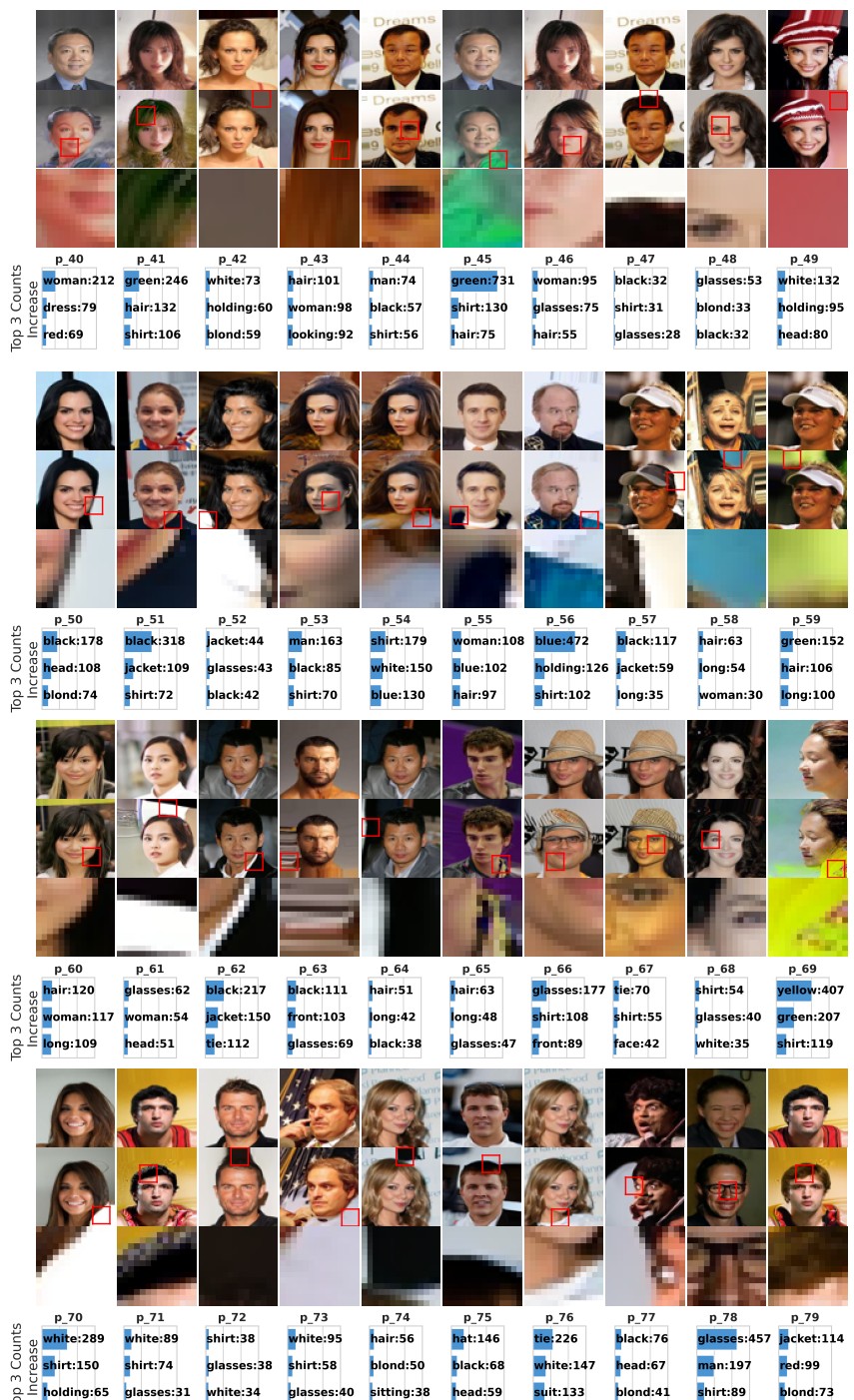

Figure 18: **Visualization of learned prototypes on CelebA with the top three increased words using BLIP for captioning (p40-p79).** Each row shows 10 prototypes with three views per prototype (top to bottom): original image, image enhanced with prototype $j$, and most activated patch (serves as the prototype visualization), and the top three increased words.

vocabulary. For instance, prototype 34 in this run corresponds to a heavy eye-makeup enhancement, but the model fails to describe it and instead produces only gender-related terms.

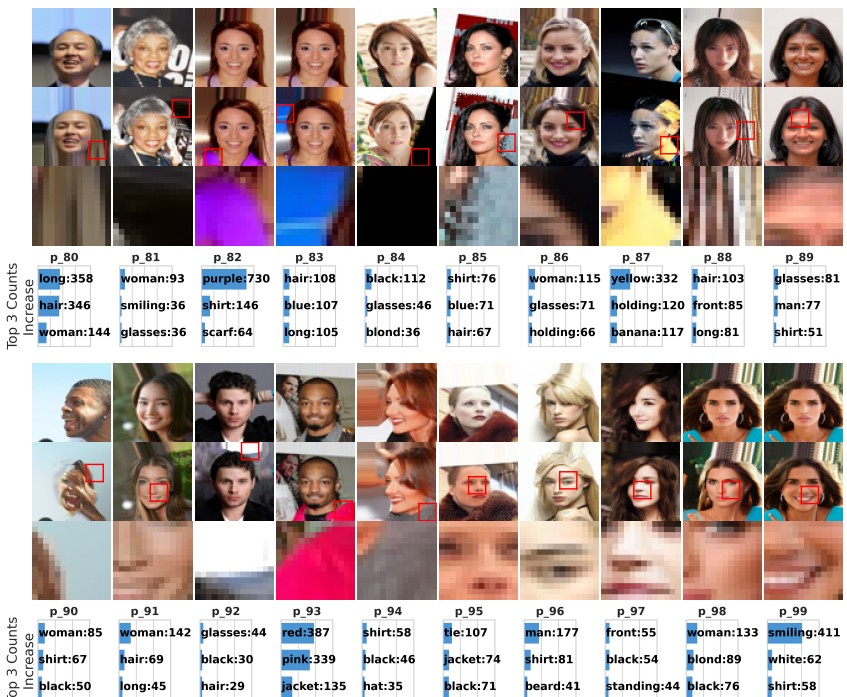

Figure 19: **Visualization of learned prototypes on CelebA with the top three increased words using BLIP for captioning (p80-p99).** Each row shows 10 prototypes with three views per prototype (top to bottom): original image, image enhanced with prototype $j$, and most activated patch (serves as the prototype visualization), and the top three increased words.

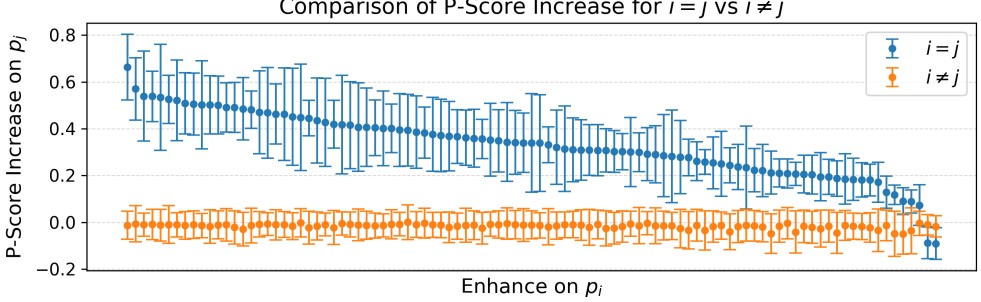

Figure 20: **Faithfulness evaluation of prototype scores.** Enhancing prototype $p_i$ leads to a clear increase in its own score ($i = j$), while scores of other prototypes remain near zero ($i \neq j$), indicating that the similarity measure behaves as intended.

## F.2 FAITHFULNESS MEASUREMENT

To evaluate the faithfulness of the prototype scores, we measure whether the score of prototype $p_i$ increases when the corresponding condition is enhanced. For each prototype $p_i$, we generate an enhanced image using $p_i$ (x-axis) and compute the change in all prototype scores (y-axis). The results are shown in Fig. 20. The target prototype shows a clear increase (mean around 0.35), whereas the remaining prototypes remain near zero, indicating that the similarity score functions as expected. The two observed outliers correspond to prototypes that encode minimal semantic information based on their visualizations.

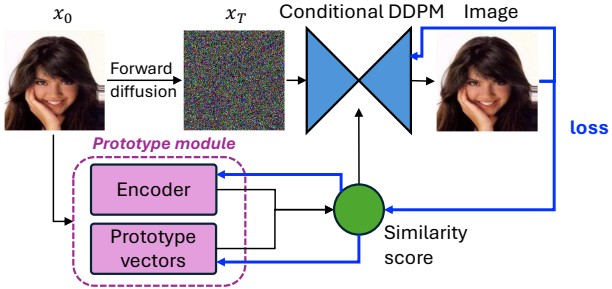

Figure 21: **Training Overview.** The black arrows show the forward computation: the encoder extracts prototypes and obtains similarity scores, which condition the DDPM during generation. The blue arrows indicate the backward path of the denoising loss, which updates both the prototype module and the conditional DDPM jointly.

## G    TRAINING STRATEGY

The training strategy for Patronus is direct and simple. All components are jointly trained with solo training objectives, i.e. the denoiser loss. Fig. 21 shows the training overview with both forward computation and backward loss propagation in black and blue arrow respectively.

## H    ROBUSTNESS OF PROTOTYPE ACTIVATION CONTROL

We evaluate the robustness of prototype activation control by comparing the cosine similarity between clean images and images generated with noise-perturbed prototype activations. We add random noise with magnitudes [0.1, 0.5, 1.0, 2.0, 5.0, 10.0]% of the maximum activation value (fixed to 2.0 in our experiments). mportantly, we perturb all prototype activations, not just a single prototype. Image similarity is measured using the cosine similarity of InceptionV3 embeddings.

Fig. 22 shows the results, clearly illustrating that the prototype activation score function is highly robust, that the cosine similarity decreases slightly at low noise levels. When the noise magnitude increase to 10%, the embedding similarity becomes comparable to that between random clean image pairs. Visual examples of generated images with noise added to the activation scores are provided in Fig. 23.

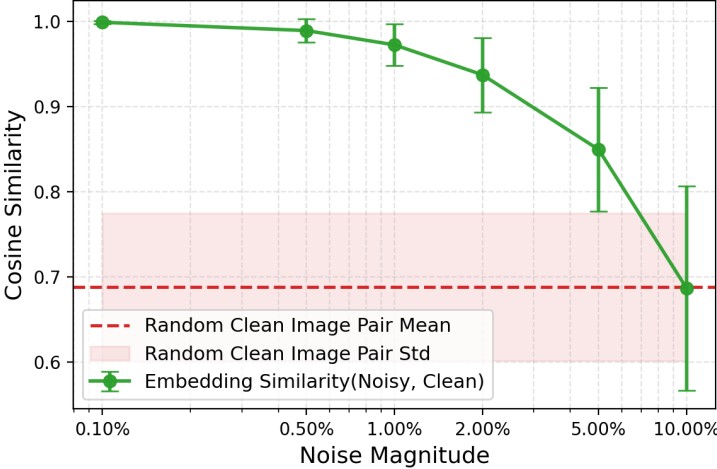

Figure 22: **Prototype activation control robustness** quantified by comparing the embedding similarity between clean images and images generated with added activation noise.

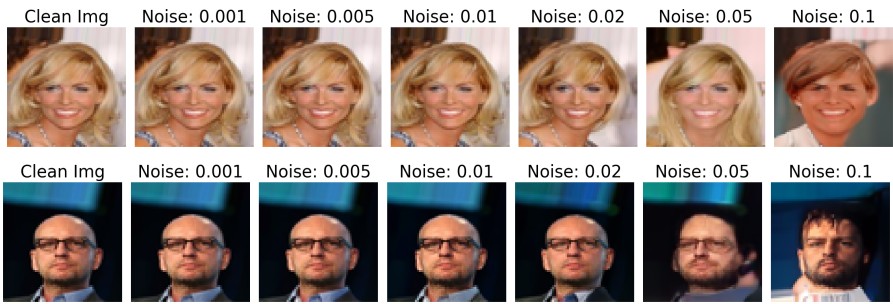

Figure 23: **Visual examples of generated images after noise is added to the activation scores.**

