# OpenReview forum: "Patronus: Interpretable Diffusion Models with Prototypes"
_ICLR.cc/2026/Conference — ICLR 2026 Poster_

### Official Review · Reviewer_hbds · 2025-10-25

**Soundness:** 3
**Presentation:** 2
**Contribution:** 3
**Rating:** 6
**Confidence:** 3

**Summary:**

This paper introduces Patronus, an interpretable diffusion model that integrates a prototypical network to encode semantic concepts within visual patches. The model learns prototypes that represent localized semantic patterns and uses their activation vectors to guide the diffusion process. Unlike post-hoc interpretability methods, Patronus embeds interpretability directly into the architecture, allowing visualization of what prototypes represent, where and when they emerge during generation, and how they interact with semantic attributes. Experiments on multiple datasets show that Patronus achieves competitive or superior generation quality and meaningful prototype disentanglement.

**Strengths:**

1. The paper introduces an architectural approach to intrinsic interpretability in diffusion models through prototype learning, differing from previous post-hoc analyses.

2. The integration between prototype activations and diffusion conditioning is mathematically consistent, with theoretical reasoning showing that adding the condition does not degrade likelihood.

3. Evaluation on many datasets (including a medical imaging dataset) demonstrates both semantic interpretability and competitive FID performance.

**Weaknesses:**

1. Some theoretical explanations are verbose and could be streamlined; a visual overview of the training pipeline would improve accessibility.

2. Joint optimization of prototype encoder, conditional DDPM, and latent diffusion may pose scalability issues for higher-resolution or text-conditioned tasks.

3. The paper acknowledges that global attributes (e.g., age or gender) are not well captured due to the patch-based encoder, but offers no structural remedy. However, I think this is also acceptable for a short-term work.

**Questions:**

1. How stable is prototype learning when scaling the model size or dataset diversity? Are prototypes consistent across training runs?

2. Can Patronus be extended to text-to-image diffusion models like Stable Diffusion, where semantics are conditioned on language rather than visual features?

3. How sensitive are prototype visualizations to noise in the prototype activation vector? Do small perturbations yield consistent semantic control?

Actually I am not an expert on diffusion theory research but a researcher on diffusion modal application, so I will check the review from other reviewers and the responses from authors to adjust my final rating.

---

> ### Author Response · Authors · 2025-11-20
> **Reply to Reviewer hbds [(1) Visual overview on training pipeline & (2) Scalability & (3) how to capture the global features & (4) Sensitivity of prototype control ]**
>
> **We really appreciate that Reviewer hbds gave positive feedback to our paper!**
>
> Here we provide the response to your concerns that are listed in the weakness and questions:
>
> **(1) An visual overview of the training pipeline (--> W1)**
>
> We do have an overview of our proposed model in Fig1.c. The main components of the proposed model patronus are: 1) a conditional DDPM; 2) prototype module that learns the prototype vectors and get prototype similarity scores as conditions.
> The training strategy of this framework is straightforward, that the whole framework is jointly trained with one loss which is the denoiser loss. We attached an extra figure in Appendix (Fig. 20),which shows the loss back propagation to the DDPM, the prototype vector and the encoder.
>
> **(2) Scalability (--> W2/Q1/Q2)**
>
> The concern from the reviewer is 2 folded:
> **a. Scalability for higher resolution**
>
> Apart from standard datasets, we have also tested on CheXpert dataset with the resolution of 224x224, which serves as evidence that our method could scale to higher resolutions and complex semantics.
>
> We would like to emphasise that:
> * The use of standard benchmark datasets is appropriate for a methodology paper. Our evaluation follows the protocol used in the most relevant work (DiffAE, InfoDiff), therefore out setup is aligned with established practice.
> * Importantly, our method is the FIRST among interpretable diffusion to additionally incorporate substantially more challenging datasets, i.e. medical imaging, providing direct evidence of scalability beyond prior work.
>
> While broader datasets could further strengthen the empirical section, this is outside the scope of a paper focused on the mechanism of a prototype-based interpretable diffusion model rather than delivering an exhaustive empirical benchmark.
>
> **b. Scalability to text-conditioned task (Q2)**
> We are not targeting text-to-image generation tasks in our work. It is possible to extend the prototype module to text-to-image diffusion models such as StableDiffusion. It might also bring interesting insights such as to certain prompts (e.g. generating a picture of a doctor), how to interpret model behavior in visual concepts (e.g. the doctor clothes, the environment, the gender features of the doctor etc). This could serve as an interesting research question for the follow-up work.
>
> However, in this work we focus more on interpreting the local visual concepts with patches (line 45-48, the key objectives of the proposed method).
>
> **(3) how to capture the global features (--> W3)**
>
> Yes, global attributes are indeed harder to capture, and this mainly comes down to how the architecture is designed. We intentionally focus on local semantics (line 48) rather than global patterns, which is different from previous methods.
> Still, it’s possible to extend the model to handle global features. For example, by adjusting the prototype encoder so each prototype can cover the whole image via larger receptive fields, or by simply adding global average pooling when computing similarity.
>
> **(4) Sensitivity of prototype control (--> Q3)**
>
> Reviewer hbds asked whether small perturbations yield consistent semantic control. We believe that the experiment on extrapolation meets the request here (Fig. 2b), where we froze the rest of the prototype score but only changed one prototype score from 0.0 to 3.0. It could be observed that the change is smooth and semantic meaningful (e.g. more and more curly hair from 0.0 to 3.0).

---

> ### Author Response · Authors · 2025-11-20
> **Reply to Reviewer hbds [(5) prototype consistency across runs]**
>
> **(5) prototype consistency across runs (--> Q2)**
>
> Thanks for pointing this out. We first emphasise that our goal is NOT to enforce consistent explanations across different models, but to explain each model’s behaviour as it is trained. If two models use different internal concepts during the generation process, then different prototypes are expected and even desirable. Nevertheless, in almost identical training conditions, i.e., the same model architecture, dataset, and objective but different random initialisations, it is still meaningful to assess whether the learned prototypes are consistent. This section evaluates that consistency.
>
> To do so, we compare prototype activation patterns on the same input batches rather than directly comparing prototype vectors (which are not aligned across runs due to arbitrary rotations in the feature space). For each batch of B images (in this case B = 512), we obtain prototype activations from both models:
>
> $A^1, A^2 ∈ ℝ^{N × B}$
>
> where each row corresponds to the activation pattern of one prototype across the batch.
> We then compute the pairwise cosine similarity matrix:
>
> $$
> S_{i,j} = \frac{\langle A^1_i, A^2_j \rangle}{\|A^1_i\| \cdot \|A^2_j\|}
> $$
>
>
> To resolve the permutation ambiguity between prototype indices, we determine the optimal one-to-one alignment using the Hungarian algorithm [1]:
>
> $$
> \pi = \arg\max_{\text{1-1 mapping}} \sum_i S_{i,\pi(i)}
> $$
>
>
> We then assess how consistently this alignment appears across all test batches.The resulting permutation consistency score is 0.882, indicating that the prototype-to-prototype mapping across seeds is largely stable, with most prototypes consistently aligned between runs. After applying the optimal alignment, the matched prototypes achieve extremely high similarity (mean ≈ 0.99), indicating that the semantic behaviour of prototypes is almost identical across seeds.
>
>
>
> [1]Kuhn, Harold W. "The Hungarian method for the assignment problem." Naval research logistics quarterly 2, no. 1‐2 (1955): 83-97.
>
>
> -----
> **We thank again for Reviewer hbds' feedback. Please reply back if there’s any further questions regarding our reply!**

---

> ### Comment · Reviewer_hbds · 2025-11-25
>
> Thanks for the authors' replies,
>
> While the authors clarified the training pipeline and gave some evidence on prototype consistency, some of my concerns remain unresolved, especially the lack of specific quantitative experimental results to support the views in the comment. In addition, I think there are still many things that need to be optimized in the layout and writing of this article. So I keep my rating for this paper.
>
> I need to explain that I am not an expert in this field, so I suggest that AC refer to the opinions of other reviewers as much as possible when deciding the results of the article.

---

> > ### Author Response · Authors · 2025-11-27
> > **Reply to Reviewer hbds: remaining concerns**
> >
> > **Thank you, we are happy that our response addressed your concerns on training pipeline and prototype consistency!**
> >
> > ## Regarding "unresolved concern the lack of specific quantitative experimental results to support the views in the comment":
> > **Could you please specify which of your comments remain unanswered? We are happy to address them.**
> >
> > You raised the following concerns that requests quantitative experimental results in the review:
> > 1. **“Q: Are prototypes consistent across training runs?”**
> > Yes, we provide quantitative experimental results for this question our rebuttal.
> > 2.  **"Q: How stable is prototype learning when scaling the model size or dataset diversity"**
> >  * **“Dataset diversity”**: Experiments on 4 different datasets (from simple to complex datasets: CIFAR10, F-MNIST, CelebA, CheXpert) should serve as good quantitative experimental results for this.
> >  * **“Model size”**: Ablation studies on Appendix B have tested on different numbers of prototypes and different prototype vector dimensions.
> > 3. **“Q: How sensitive are prototype visualizations to noise in the prototype activation vector? Do small perturbations yield consistent semantic control?”**
> >
> > We assess the sensitivity of prototype visualisation in 2 ways:
> > * **Qualitatively**, it could be analysed by the extrapolation experiments done in fig. 2b via the smooth and semantic change on generated images. Besides, the interpolation experiments (fig. 2c) are also a good proof that interpolating the conditions leads to correspondingly smooth transitions in the outputs.
> > * **Quantitatively**, we added an experiment to assess the robustness of prototype activation control by comparing the cosine similarity between clean images and images generated with noise-perturbed prototype activations.We injected random noise at magnitudes [0.1, 0.5, 1.0, 2.0, 5.0, 10.0]% of the maximum activation value (fixed to 2.0). Similarity was measured using the *cosine similarity of InceptionV3 embeddings*.
> >    The results are:
> > | Noise Level (%) | Cosine Similarity (mean ± std) |
> > |-----------------|--------------------------------|
> > | 0.1             | 0.9991 ± 0.0016                |
> > | 0.5             | 0.99 ± 0.01                |
> > | 1.0             | 0.97 ± 0.02                |
> > | 2.0             | 0.94 ± 0.04               |
> > | 5.0             | 0.85 ± 0.07                |
> > | 10.0            | 0.69 ± 0.12                |
> >
> > These results demonstrate that the **prototype activation score function is highly robust to activation perturbations.**
> > At the 10% noise level, the similarity becomes comparable to that between random pairs of clean images, indicating that the perturbation is sufficiently large to break the alignment at 10% perturbation.
> > **Figures 21–22 in Appendix H** present the corresponding plots as well as visual examples of generated images under different noise levels.
> >
> > We don’t understand which other comments should be answered via experimental results, but would be happy to improve – **could you please clarify?** Thank you!!
> >
> > ## Regarding the comment on “layout and the writing could be optimized”
> > Reviewer hbds mentioned the writing could be optimized and lowered the presentation score to “poor”, which seems odd given that reviewer hbds’ concrete doesn’t really comment on which part of the article they find poorly presented. This would be helpful for revising our paper.
> >
> > ------
> > We thank again for your feedback, since it is still during the discussion phase, we would very much like to hear back from you to clear all concerns!

---

> ### Comment · Reviewer_hbds · 2025-11-27
>
> Thanks for the authors' reply again.
>
> I apologize for not explaining clearly why I adjusted the score of the presentation. In fact, I want to remind you that some of the image sizes and font sizes in the visualization parts of the revised paper are too small to read, but I forgot to add the new Comment when submitting the modification.
>
> The additional experiments were very good, which further alleviated my concerns. I will adjust the rating after the author completes the revisions.

---

> > ### Comment · Reviewer_hbds · 2025-11-27
> >
> > In addition, I would like to remind the authors that there are some errors in the serial number of the figures of the paper. Don't forget to fix this issue.

---

> > ### Author Response · Authors · 2025-11-27
> > **Reply to hbds: Thank you for acknowledge additional experiment results; We revised the image size and font sizes.**
> >
> > **Thank you! For replying and letting us know that the additional experiments further alleviated your concerns!**
> >
> > ## Regarding the presentation of the paper:
> > * We enlarged the image and font size of all figures to make them more print-friendly and easier to read. We believe they look better now! Please let us know if any figure is still not satisfactory to you.
> >
> > ## About the "errors in the serial number of the figures of the paper":
> > * We went through the paper and ensured that the figure numbers follow the order in which they are referenced in the text. This may result in some figures appearing earlier than figures with smaller numbers, due to LaTeX’s optimized layout. If there is a specific ordering issue you notice, please let us know! Really appreciate it!
> >
> > **Our revised paper is now uploaded to openreview. Please let us know if there's any remaining concerns.**
> > Thank you!

---

> > > ### Comment · Reviewer_hbds · 2025-11-28
> > >
> > > Thank you for the author's proactive response. It looks much better now. It seems that the OpenReview system cannot edit rating. I will increase my rating when it is possible to edit scores. If this edit button no longer opens, I hope to express to AC that I have no objections to the presentation of this paper.

---

### Official Review · Reviewer_o2Si · 2025-10-26

**Soundness:** 3
**Presentation:** 3
**Contribution:** 2
**Rating:** 4
**Confidence:** 4

**Summary:**

This paper introduces "Patronus," a new type of diffusion model that tries to open up the "black box" of how these models generate images. The core idea is to build a "prototypical network" directly into the U-Net. This network learns a set of visual patterns (prototypes) from the data. As the model denoises an image, it maps its internal features to these prototypes, letting us see what patterns are being used, where they're being placed, and when (at which timestep) they appear. The authors show this can be used to understand the model's logic, catch it "cheating" by learning bad correlations (shortcut learning), and track how concepts form from noise.

**Strengths:**

Overall, the work is interesting.

1.	The core idea is highly original. We’ve seen prototypes used for interpretability in classification, but applying this idea to understand the internal dynamics of a diffusion model as it generates is a very creative and insightful leap.

2.	The ability to detect "shortcut learning" is a particularly high-impact claim, especially for the medical imaging applications they explore, where you absolutely cannot have a model focusing on the wrong artifacts.

**Weaknesses:**

I have a few key reservations that kept me from being completely convinced.

1.	My main concern is about the classic "interpretability vs. performance" trade-off. The authors claim "strong generative performance," but adding a whole prototype network inside the U-Net can't be computationally free. I really need to see a clear analysis of the cost. What’s the hit to the FID score compared to a standard baseline with the same parameter count? What's the extra latency during inference? This needs to be quantified.

2.	I'm also worried about scalability. The experiments mentioned seem to be on smaller-scale datasets (like CIFAR or low-res medical images). I'm skeptical that this method, as-is, would scale to high-resolution (e.g., 256x256 or 512x512) and high-complexity (e.g., ImageNet) datasets. The computational cost of computing patch-prototype similarity at every single step for large batches and high-res feature maps seems like it would be a major bottleneck.

3.	The entire utility of this method hinges on the quality of the learned prototypes. The paper needs to do more to convince me that these prototypes are (a) genuinely meaningful to a human, (b) diverse and not just "collapsed" to a few simple textures (even though the authors directly use a loss function to make it diverse), and (c) actually represent what the model is really thinking. Qualitative examples can be cherry-picked, so I’m looking for a more quantitative analysis of prototype quality.

4.	This method is only tested on unconditional diffusion models (here we do not treat the prototype activation vector as ‘real’ conditions). Could this method be applied to text/image conditioned diffusion models as well?

**Questions:**

To help clarify these points, I have a few questions for the authors:

1.	**The Performance Trade-off**: Could you provide a direct, head-to-head comparison of generative quality (FID, etc.) and computational cost (GFLOPs, wall-clock time) between Patronus and a standard diffusion baseline of similar size?

2.	**The "Shortcut Learning" Claim**: This is a big selling point. Could you walk me through more concrete examples from your experiments (maybe from the medical dataset) where Patronus clearly identified an unwanted correlation that a baseline model was exploiting?

3.	**Scaling Up**: What are the real bottlenecks in scaling Patronus to something like ImageNet? Have you thought about or explored any approximations (e.g., to the prototype matching step) to make it more efficient at high resolutions?

---

> ### Author Response · Authors · 2025-11-20
> **Reply to Reviewer o2Si [ (1)Quantitative analysis on interpretability vs. performance trade-off & (2) Scalability ]**
>
> **We’d like to genuinely thank Reviewer o2Si for your feedback, especially for your acknowledgement of the originality of our work, and the important impact of patronus capability of detecting shortcuts from its interpretability. As the reviewer notes, digging into how the diffusion model works, and aligning with human understanding and values, are definitely our motivations for this work.**
>
> Here we provide the response to your concerns that are listed in the weakness and questions:
>
> ----
> **(1) Quantitative analysis on interpretability vs. performance trade-off ( →W1 & Q1)**
>
> We hereby provide the empirical stats of the computational cost of adding the prototype module.
>
> **a. Computational cost in training**:
>
> We have performed some measurements on model size and performance, which you can see in table below:
>
> Tab.1 *Comparison between patronus and a plain Unet: “Interpretability v.s. Performance”*
>
> | Model                                           | Interpretable? | Params   | FID                 | GFLOPs / forward | Wall-clock time (64 imgs) |
> |-------------------------------------------------|----------------|----------|----------------------|------------------|-----------------------------|
> | plain U-Net                                      | NO             | 25.02M   | 8.1274 ± 0.035       | 536.2921         | 71.17 ± 2.42                |
> | Patronus (U-Net + prototype module, #p = 64)    | YES             | 25.24M   | 6.5395 ± 0.0676      | 536.3634         | 75.46 ± 1.63                |
> | Patronus (U-Net + prototype module, #p = 128)   | YES           | 25.27M   | 5.1264 ± 0.0488      | 536.3645         | 75.14 ± 2.43                |
>
> *Other shared hyper parameters*:
> Lr=1e-4, epochs = 200, Input resolution: 64x64, dataset: CelebA.
> Unet structure: 4-stage U-Net, C=64×(1,2,4,4), 2 ResBlocks each, GN+SiLU, no attention, sinusoidal time embedding.
> Prototype encoder structure: A 4-layer Conv–ReLU encoder with channels 1→32→64→64→128
> (kernel = 3, strides [2,1,1,1], paddings [1,0,0,0]). shape of p: (1,1,128).
>
> As shown above, because the prototype module has far fewer parameters than the U-Net, the total number of trainable parameters remains almost unchanged with or without the prototype module. The FID also improves after adding the prototype module. We further report the GFLOPs per forward pass, where the difference between models is negligible.
> Apart from that, we have more ablation study on #p and the shape of p on Appendix B, which could also indicate the interpretability and performance trade-off in some way, as more prototypes, in principle should increase the interpretability until it collapses.
>
> **b. Computational cost during inference**
>
> Thank you for this important note. We empirically collect the time (in seconds) needed for 64 images (one batch) for the 3 models in Tab 1. It shows that the latency for inference is quite small.
>
> ----
> **(2) Scalability (→W2 & Q3)**
>
> a. Reviewer asked: “Whether the method could scale to high resolution and high-complexity datasets”.
>
> Apart from standard datasets, we have also tested on CheXpert dataset with the resolution of 224x224, which serves as evidence that our method could scale to higher resolutions and complex semantics.
>
> We would like to emphasise that:
> * The use of standard benchmark datasets is appropriate for a methodology paper. Our evaluation follows the protocol used in the most relevant work (DiffAE, InfoDiff), therefore out setup is aligned with established practice.
> * Importantly, our method is the FIRST among interpretable diffusion to additionally incorporate substantially more challenging datasets, i.e. medical imaging, providing direct evidence of scalability beyond prior work.
>
> While broader datasets could further strengthen the empirical section, this is outside the scope of a paper focused on the mechanism of a prototype-based interpretable diffusion model rather than delivering an exhaustive empirical benchmark.
>
> b. Reviewer: “The computational cost of computing patch-prototype similarity at every single step for large batches and high-res feature maps seems like it would be a major bottleneck”
>
> We here show that getting the similarity score of prototypes at every step is not costly at all.
> * For training, the prototype similarity score only needs to be computed once with the computation cost of 6.15 GFLOPs. Whereas, predicting the noise at EACH timestep takes  536.36 GFLOPs. For a 1000 timestep diffusion model, the prototype similarity computation part only takes around 0.001% of time.
> * For inference. If the condition is given, then it takes nothing to compute the similarity score. If not, we use a separate 1D Unet to sample the similarity score, with the computational cost around 14.96 GFLOPs per step. We applied a 1D Unet for 100 timestep, therefore the computation cost will be around 0.28% of the total time for generating one sample.
>
> Therefore, the overall cost of the similarity computations is negligible in practice.

---

> ### Author Response · Authors · 2025-11-20
> **Reply to Reviewer o2Si [(3) Evaluation of the interpretability of patronus ]**
>
> **(3) Evaluation of the interpretability of patronus (--> W3)**
>
> Reviewer o2Si requested more evidence that:
>
> **a. “The learned prototypes are genuinely meaningful to a human”.**
>
> In our paper, we have provided evidence that the learned prototypes are semantically meaningful by reconstraction / Interpolation/ visualizing the prototypes/ manipulation and extrapolation in sec 4.1. In XAI community, especially for interpretable diffusion literatures (e.g. DiffAE and InfoDiff), those are typical ways to validate that prototypes contain semantics that align with human understanding.
>
> Upon Reviewer o2Si’s request for more quantitative evaluation, we did an extra experiment on exploring whether we can use languages to prove the alignments of the visual concepts and human understandings. Specifically, we like to address if our prototypes encode visual features that are visible and understandable via human concepts. For that, we applied the BLIP model (Salesforce/blip-image-captioning-large at huggingface) to caption the enhanced image by prototype j and the original image, then we compared which word has a significant increase. For example, for the prototype which represents “curly hair” by visualization (Fig.3a), the top 3 words with highest increase after the prototype modification are: {‘curly’: 409, ‘hair’:307, ‘long’: 66}, the value is the word frequency with enhanced p_j minus the word frequency for the original image in 672 samples.
> We also provide the whole list of the top 3 increased words together with the visualization of the prototypes of one run in Appendix F.1 (Fig.17-19).
>
> **b. “The learned prototypes are diverse and not just collapsed to a few simple textures”**
>
> In our paper, we have already shown that: applying a Prototype Distinct Loss to encourage prototype disentanglement to the baseline patronus does not cause substantial change in the learned prototypes. The quantitative analysis over this is on Appendix D.1, where we showed the distribution of Pearson correlation of prototype pairs with the baseline model, baseline model+forcing prototype disentanglement. From the distribution we can also see that most of the pairs have correlation score close to 0.0, which indicates that they are diverse. The qualitative analysis of such could be found by fig 11-12 in appendix, where we showcase the full list of prototype visualization of one run.
>
> **c.  “The learned prototypes actually represent what the model is really thinking. ”**
>
> We believe this request is most close to the faithfulness measurement for interpretability evaluation. “faithfulness” refers to how accurately it reflects the true reasoning process of the model [1,2].
>
> Upon Reviewer o2Si’s request for more quantitative evaluation, we did an extra experiment to measure the faithfulness of our model, by validating whether the p score increases when enhancing it with the condition.
> The results are shown in the following figure [if we cannot reply with images - put it in appendix ]
> As we can not attach figures here, we attached the result in appendix (Fig.20).
> On x-axis, we enhanced the prototype p_i and generate a new image, on y-axis we compute the p score increasement on all prototypes, we can see from this figure that almost all prototypes are significantly increased  (mean increment ~0.35) compared to other prototypes (which basically remains unchanged with the mean ~0.0) after the enhancement, which show that the similarity score is functioning as expected.
> The 2 outliers are also interesting to see. After investigating, we believe that those 2 prototypes encode very little semantic information, from their visualization.
>
> [1]Wiegreffe, Sarah, and Yuval Pinter. "Attention is not not explanation." arXiv preprint arXiv:1908.04626 (2019).
>
> [2]Herman, Bernease. "The promise and peril of human evaluation for model interpretability." arXiv preprint arXiv:1711.07414 (2017).

---

> ### Author Response · Authors · 2025-11-20
> **Reply to Reviewer o2Si [(4) potential to extend to StableDiffusion & (5) A walk through of: Patronus helping detecting unwanted correlations ]**
>
> **(4) potential to extend to StableDiffusion (--> W4)**
>
> It is possible to extend the prototype module to a text-to-image diffusion model like StableDiffusion. [3] introduced an extra neural network called ControlNet to add spatial conditioning to the text-to-image diffusion model. Thus, adding extra conditions is possible. It might also bring interesting insights such as to certain prompts (e.g. generating a picture of a doctor), how to interpret model behavior in visual concepts (e.g. the doctor clothes, the environment, the gender features of the doctor etc). This could serve as an interesting research question for the follow-up work.
>
> [3] Zhang, Lvmin, Anyi Rao, and Maneesh Agrawala. "Adding conditional control to text-to-image diffusion models." In Proceedings of the IEEE/CVF international conference on computer vision, pp. 3836-3847. 2023.
>
> ----
>
> **(5) A walk through of: Patronus helping detecting unwanted correlations (--> Question 2)**
>
> *Example 1*: We have a case with introduced bias in our paper (sec 4.4), where we train the patronus with unwanted correlation between hair color and smile in CelebA: that all blonde/brown-haired images smile, while black-haired ones do not. As shown in Fig.4a, the concept of hair color and smiling is then entangled in prototype vectors, that if you enhance the concept of “black hair”, the generated image has less smile; whereas enhancing “blonde hair”, there is a bigger smile. We have also quantitatively analyse the effect in appendix (sec C.1 and fig.10), where we first find the most representative prototypes to “black hair”, “blonde hair”, and “smile”, and compute the Spearman correlation between them, which shows that the lighter hair color is more positively correlated with smiles.
>
> *Example 2*: In Fig.2b, when explorating the prototype “eye make-up”, we can observe that the female features are decreasing together with the “eye make-up” features. More interestingly, the feature of having glasses seems to be highly negatively correlated with “eye make up” features too.
>
>
> ----
>
> **We thank again for Reviewer o2Si’s feedback. Please reply back if there’s any further questions regarding our reply!**

---

> ### Author Response · Authors · 2025-11-27
> **To Reviewer o2Si: We would very much like to hear whether our reply solve your concern!**
>
> Dear Reviewer o2Si, we have responded to your concerns, and have not heard back from you **since last Thursday (7 days ago)**.
>
> **We would very much like to hear whether our reply solve your concern!** If not, please let us know, we will make sure to resolve all your concerns before the discussion phase ends.
>
> Thank you!!

---

### Official Review · Reviewer_WfoU · 2025-10-30

**Soundness:** 3
**Presentation:** 3
**Contribution:** 2
**Rating:** 4
**Confidence:** 3

**Summary:**

This paper introduces Patronus, a diffusion model that integrates a prototypical network within the denoising architecture to provide intrinsic interpretability.
Instead of relying on post-hoc feature visualization or external semantic encoders, Patronus learns patch-based prototypes that encode localized semantic patterns (''what"), their spatial location ("where"), and temporal emergence during denoising ("when").
Prototype activation vectors condition the diffusion process, allowing direct visualization, manipulation, and bias diagnosis (i.e., shortcut learning).
Empirical results across five datasets:
- Achieves competitive or superior FID and latent quality (AUROC/TAD)
- Produces interpretable prototype semantics
- Enables diagnosis of unwanted correlations (e.g., hair color - smile);
- Reveals temporal emergence of visual features.

**Strengths:**

- Combines a prototypical encoder and a conditional diffusion process, embedding interpretability directly rather than via post-hoc probing.
- "What, where, when" decomposition is compelling and concretely supported by prototype visualizations and activation dynamics.
- Prototypes correspond to semantically meaningful attributes (e.g., smile, hair color, collar) and can be manipulated for controllable image editing.
- Demonstration of shortcut learning (hair color ↔ smile) is convincing and highlights societal relevance.
- Ablations explore prototype distinctness, disentanglement, and conditional generation.
- Figures and examples are intuitive and make the technical ideas clear.

**Weaknesses:**

- The prototype extraction is largely adapted from ProtoPNet, with limited theoretical or algorithmic innovation in how prototypes are integrated beyond being used as conditional guidance.
- The interpretability claims rely mostly on qualitative visualization; no quantitative evaluation (e.g., localization accuracy, faithfulness, or human interpretability studies) is provided.
- While visualizations are appealing, there is no test showing that manipulating a prototype truly corresponds to causal semantic change rather than correlated artifacts.
- It’s unclear how many prototypes are needed, whether semantics are stable across seeds, or how sensitive the results are to prototype dimensionality.
- Comparison is limited to DiffAE and InfoDiff. More modern interpretable or controllable diffusion baselines (e.g., DDPM inversion [1]) could be considered.
- The paper could better position itself relative to prior encoder-conditioned approaches. What specifically differentiates the prototype activations from lower-dimensional semantic vectors used in DiffAE-like methods.
- Theoretical claims in Section 3.5 (showing conditioning "cannot degrade" the ELBO) are informal and would benefit from a more rigorous derivation or empirical validation.

[1] Huberman-Spiegelglas et al., CVPR’24.

**Questions:**

1. How stable are the prototypes across random seeds? Do the same semantic concepts consistently emerge?
2. Can the authors quantify interpretability (e.g., localization or faithfulness metrics)?
3. How does Patronus compare to direction-based interpretability methods (e.g., PCA directions, linear semantic editing)?
4. Beyond hair - smile, does it detect other spurious links (e.g., makeup - gender cues)?
5. How do performance and disentanglement vary with number of prototypes and prototype dim?
6. Could the prototype activation vector be used for cross-domain transfer or editing (e.g., applying "smile" from one dataset to another)?

---

> ### Author Response · Authors · 2025-11-20
> **Reply to Reviewer WfoU [(1)Concerns about novelty & (2) quantitative evaluation over the interpretability & (3) Quantitative evaluation on image editing & (4) Ablation study on prototypes]**
>
> **We thank reviewer WfoU for your feedback! Especially the positive remarks regarding the “what–where–when” interpretability of our model, its potential societal relevance in addressing shortcut learning, and the completeness of our ablation study.**
>
> Here we provide the response to your concerns that are listed in the weakness and questions:
>
> -----
>
> **(1) Does the closeness to ProtoPNet lead to lack of novelty ? (W1)**
>
> Regarding the concern that integrating prototype extraction from ProtoPNet may limit the novelty of our contribution, we respectfully disagree.
> While the prototype extraction component is indeed inspired by ProtoPNet, this does not diminish the novelty of the paper.
> In our case, the integration is not a direct reuse: prior work in interpretable diffusion model conditions on the encoder output, whereas we use prototype similarity scores as the conditioning signal. This substantially reduces the dimensionality required for diffusion guidance while retaining the semantic capacity of prototype features. We have discussed this distinction in the Related Work section (lines 129–139).
>
> ----
>
> **(2) More quantitative evaluation over the interpretability (--> W2&Q2)**
>
> We hereby added more quantitative evaluation of the interpretability:
>
> **a. Language alignment of the learned prototypes:**
>
> We did an extra experiment on exploring whether we can use languages to prove the alignments of the visual concepts and human understandings. Specifically, we applied the BLIP model (Salesforce/blip-image-captioning-large in huggingface) to caption the enhanced image by prototype j and the original image, then we compared which word has a significant increase. For example, for the prototype which represents “curly hair” by visualization (Fig.3a), the top 3 words with increasement are: *{‘curly’: 409, ‘hair’:307, ‘long’: 66}*, the value is the word frequency with enhanced p_j minus the word frequency for the original image in 672 samples. We also provide the whole list of the top 5 increased words together with the visualization of the prototypes of one run in Appendix F.1 (Fig.17-19)
>
> **b. Faithfulness:**
>
> “Faithfulness” refers to how accurately it reflects the true reasoning process of the model [1,2].
>
> We did an extra experiment to measure the faithfulness of our model, by validating whether the p score increases when enhancing it with the condition. The results are shown in the following figure.
> As we can not attach figures here, we attached the result in appendix (Fig.20).
>
> On x-axis, we enhanced the prototype p_i and generate a new image, on y-axis we compute the p score increasement on all prototypes, we can see from this figure that almost all prototypes are significantly increased compared to other prototypes (which basically remains unchanged with the mean ~0.0) after the enhancement, which proved that the similarity score is functioning in the way that it suppose to do.
> The 2 outliers are also interesting to see. After investigating, we believe that those 2 prototypes encode very little semantic information, from their visualization.
>
> Due to the time limit, we can not provide human interpretability studies at this point.
>
> [1]Wiegreffe, Sarah, and Yuval Pinter. "Attention is not not explanation." arXiv preprint arXiv:1908.04626 (2019).
> [2]Herman, Bernease. "The promise and peril of human evaluation for model interpretability." arXiv preprint arXiv:1711.07414 (2017).
>
> ------
> **(3) Quantitative evaluation on image editing (W3)**
>
> Reviewer WfoU mentioned: “no test showing that manipulating a prototype truly corresponds to causal semantic change rather than correlated artifacts.”
>
> The evaluation with language model, partially proved that after enhancing the image with prototypes, the semantic changes.
> However, we would like to emphasise that Patronus is not a precise image editing tool, it is an interpretable tool. It offers a way to visualise what visual concepts have been learned in your diffusion model, whether they are entangled or biassed, or whether there are some important concepts missing.
>
> -------
> **(4) Ablation study on prototypes (W4 &Q5)**
>
> It’s a good point, and we DO have those ablation studies in our appendix.
> Reviewer WfoU asked:
> * “It’s unclear how many prototypes are needed”: We do have ablation study on the number of prototypes in appendix B.1.
> * “how sensitive the results are to prototype dimensionality”: We do also have an ablation study on prototype dimensionality in appendix B.2.

---

> ### Author Response · Authors · 2025-11-20
> **Reply to Reviewer WfoU [(5)prototype consistency across seed & (6) Comparison to DDPM inversion & (7) Positioning relative to encoder-conditioned approaches ]**
>
> **(5) prototype consistency across seed (W4 & Q1)**
>
> Thanks for pointing this out. We first emphasise that our goal is NOT to enforce consistent explanations across different models, but to explain each model’s behaviour as it is trained. If two models use different internal concepts during the generation process, then different prototypes are expected and even desirable. Nevertheless, in almost identical training conditions, i.e., the same model architecture, dataset, and objective but different random initialisations, it is still meaningful to assess whether the learned prototypes are consistent. This section evaluates that consistency.
> To do so, we compare prototype activation patterns on the same input batches rather than directly comparing prototype vectors (which are not aligned across runs due to arbitrary rotations in the feature space). For each batch of B images (in this case B = 512), we obtain prototype activations from both models:
> $$
> A^1, A^2 \in \mathbb{R}^{N \times B}
> $$
>
> where each row corresponds to the activation pattern of one prototype across the batch.
> We then compute the pairwise cosine similarity matrix:
> $$
> S_{i,j} = \frac{\langle A^1_i, A^2_j \rangle}{\|A^1_i\| \cdot \|A^2_j\|}
> $$
>
> To resolve the permutation ambiguity between prototype indices, we determine the optimal one-to-one alignment using the Hungarian algorithm [1]:
> $$
> \pi = \arg\max_{\text{1-1 mapping}} \sum_i S_{i,\pi(i)}
> $$
>
> We then assess how consistently this alignment appears across all test batches.The resulting permutation consistency score is 0.882, indicating that the prototype-to-prototype mapping across seeds is largely stable, with most prototypes consistently aligned between runs.After applying the optimal alignment, the matched prototypes achieve extremely high similarity (mean ≈ 0.99), indicating that the semantic behaviour of prototypes is almost identical across seeds.
>
>
>
> [1]Kuhn, Harold W. "The Hungarian method for the assignment problem." Naval research logistics quarterly 2, no. 1‐2 (1955): 83-97.
>
> ------
>
>
> **(6) Comparison to DDPM inversion (W5)**
>
> DDPM inversion is a work that aims at image editing.
> Interpretable Diffusion models and diffusion for image editing are two different tasks. In other words, image editing tasks focus more on how to precisely edit an image, which often requires extra given annotation or information about what you would like the image to edit to.
> In contrast, Patronus is trained in a completely unsupervised manner. In interpretable diffusion models, we edit the image to only show that the interoperability is solid, to prove that the semantics are meaningful. Therefore it is very hard to put comparison in between those works as the goal is fundamentally different.
> But since Reviewer WfoU pointed this out, we see the possible confusion it might cause. We will consider adding another paragraph in our related work or in appendix to emphasis the difference of these two types of tasks.
>
> ------
> **(7)  We already position Patronus relative to encoder-conditioned approaches (W6&Q3)**
>
> Reviewer mentioned: “The paper could better position itself relative to prior encoder-conditioned approaches,”
>
> * “The paper could better position itself relative to prior encoder-conditioned approaches”: This is exactly what we have done in our paper. In sec 2.2 (related work: interpretable diffusion model) we categorised the current literatures into 2 parts, a. Semantic Interpretation of Internal Features and b. Autoencoder-based semantic feature extraction for guidance. We mentioned Patonus is closer to the second type, as we aim at building intrinsic interpretable models rather than post-hoc methods.
> * “What specifically differentiates the prototype activations from lower-dimensional semantic vectors used in DiffAE-like methods”: in line 135-139: we compared how patronus is different from other auroencoder based framework by emphasising
> * >  [from our manuscript] “While these methods extract global semantic features, Patronus instead captures local features through a prototypical network. Another key distinction lies in how the diffusion model is guided: Rather than direct semantic information, we use the prototype activation vector. This approach significantly reduces the dimensionality needed for diffusion guidance while preserving enough capacity in prototype features to encode the same semantic information.”
>
> This is also related to the third question reviewer WfoU raised about “How does Patronus compare to direction-based interpretability methods (e.g., PCA directions, linear semantic editing)”, as this part of literature lies in “a. Semantic Interpretation of Internal Features”. We did not provide experiments regarding those as patronus are more closer to the autoencoder guidance part of the literature.

---

> ### Author Response · Authors · 2025-11-20
> **Reply to Reviewer WfoU [(8) Concerns related to the theoretical claims & (9) Reply to Q4 & (10) Reply to Q6 ]**
>
> **(8) Concerns related to the theoretical claims in sec 3.5 (W7)**
>
> We improved the proposition and proof in the manuscript accordingly, please check the new sec 3.5. We hope this is more clear compared to the previous version.
>
> ----
> **(9) Q4: Beyond hair - smile, does it detect other spurious links (e.g., makeup - gender cues)?**
>
> Yes exactly. In the paper, we mentioned in line 416, that ”Similar patterns emerge without subset manipulation. In Fig. 2b, when decreasing eye makeup, the female feature is decreasing, and the glasses attribute is increasing.
>
> ----
> **(10) Q6: Could the prototype activation vector be used for cross-domain transfer or editing (e.g., applying "smile" from one dataset to another)?**
>
> The reviewer WfoU asked whether prototype activations extracted from one domain (using encoder E1, E2​) could be used to guide generation in another domain (using generator G1, G2​). This is an interesting direction, and our current formulation does not explicitly target cross-domain transfer.
> Conceptually, such transferability would depend on the degree of semantic overlap between the two datasets. If the domains share similar underlying structure, for instance, two human-face datasets, then certain prototype activations (e.g., “smile”) may indeed remain meaningful across encoders. In such cases, cross-domain editing could be possible in principle. However, if the domains differ substantially, prototype semantics may no longer align, and transfer would be considerably less reliable.
> While we agree this is a promising avenue for future work, it is beyond the scope of the present paper, which focuses on understanding and evaluating prototype-based interpretability within a single domain. We therefore consider this question orthogonal to the main contributions, but we appreciate the reviewer for highlighting this potential extension.
>
>
> ------
> **We thank again for Reviewer WfoU's feedback. Please reply back if there’s any further questions regarding our reply!**

---

> ### Author Response · Authors · 2025-11-27
> **To Reviewer WfoU: We would very much like to hear whether our reply solve your concern**
>
> Dear Reviewer WfoU, we have responded to your concerns, and have not heard back from you **since last Thursday (7 days ago)**.
>
> **We would very much like to hear whether our reply solve your concern!** If not, please let us know, we will make sure to resolve all your concerns before the discussion phase ends.
>
> Thank you!!

---

### Official Review · Reviewer_JUEx · 2025-10-30

**Soundness:** 3
**Presentation:** 2
**Contribution:** 3
**Rating:** 4
**Confidence:** 3

**Summary:**

This paper introduces Patronus, a diffusion model that incorporates prototypes to provide more interpretable generation. The model learns patch-level prototypes and uses prototype activation vectors to condition the diffusion process. The authors claim that this enables understanding of what semantic patterns emerge, and where and when they appear during denoising. The paper also proposes a sampling-based prototype visualization strategy. Experiments on several datasets are presented, focusing on semantics, reconstruction, guidance, and bias analysis.
The goal of embedding interpretability into diffusion models is timely and important. The direction of prototype-based interpretability for diffusion is interesting.

**Strengths:**

* Tackles an important problem: interpretability in diffusion models.
* Clear motivation for moving toward intrinsic interpretability rather than post-hoc analysis.
* Prototype idea has conceptual appeal and builds on a known line of interpretable ML research.
* Good results in certain controlled settings with simple datasets.

**Weaknesses:**

* **Clarity issues:** The paper is written in a cryptic way and often hides a simple idea behind heavy verbal descriptions. The prototype module is not sufficiently described at the engineering and procedural level. For example, it remains unclear whether the prototype encoder is a standard CNN, how it is trained relative to the denoiser, and whether prototypes are learned or chosen a priori. Section 3.1 in particular does not provide essential clarity on the prototype encoder architecture and training scheme.
* **Name inconsistency:** This is actually a minor comment, but the acronym “Prototype-Assisted TRANsparent diffuSion model” does not actually map to “Patronus”.
* **Proposition lacks rigor:** The stated proposition is informal and the proof is neither formal nor convincing. There is no precise mathematical statement or measurable criterion. The argument reads as a loose justification rather than a proof. It should be rewritten to be mathematically precise or removed.
* **Reverse DDIM reference is vague:** The paper refers to a “reverse DDIM process” (line 313) but never clearly defines how it is implemented.
* **Weak empirical depth:** Experiments rely mainly on small and simple datasets (FMNIST, CIFAR-10, CelebA subsets). This limits confidence in claims about semantic interpretability and scalability. Higher-resolution and more complex datasets (ImageNet, large natural scenes) would significantly strengthen the case.
* **Interpretability not fully grounded:** The prototype activations are visually inspected and interpreted manually. There is no systematic evaluation of interpretability quality, faithfulness, or prototype consistency beyond selected qualitative examples.
* **Simple baseline missing:** A naive “mask-based or patch-embedding supervision” or a direct attention-map visualization baseline would help demonstrate that prototypes add more than a different form of patch scoring.

**Questions:**

1. What exactly is the architecture of the prototype encoder? Is it a ResNet? How many layers? Is it trained jointly or frozen after pretraining?
2. Are prototypes randomly initialized and learned, or do you initialize from real patches?
3. How is the reverse DDIM performed in practice? Please provide pseudocode or a precise description.
4. Try to be more clear, direct, and less cryptic in the explanation.

---

> ### Author Response · Authors · 2025-11-20
> **Reply to Reviewer JUEx [(1)Details about prototype encoder and training strategies & (2)Name inconsistency & (3)Proposition lacks rigor & (4)Reverse DDIM reference]**
>
> **We really appreciate Reviewer JUEx’s feedback to our paper. We are glad that you see the contribution in integrating prototypes into diffusion models, and the motivation with intrinsic interpretability.**
>
> Here we provide the response to your concerns that are listed in the weakness and questions:
>
> **(1) Details about prototype encoder and training strategies (W1&Q1)**
>
> * “The prototype module is not sufficiently described”: The details of the prototype module are described in Sec 3.1 together with fig.1c-bottom.
> * “it remains unclear whether the prototype encoder is a standard CNN”: The encoder is indeed a CNN, which is sort of mentioned by line 156, when specifying how we mapping the output neural to the patch in the input imaging using the receptive field.
>     * In principle, this encoder could be any kind of CNN, as long as receptive fields are retrievable.
>     * The specific structure of the encoder used in this work is: A 4-layer Conv–ReLU encoder with channels 1→32→64→64→128 (with kernel = 3, strides [2,1,1,1], paddings [1,0,0,0]). shape of p: (1,1,128).
>     * We will add this information into the paper.
>
> * “how it is trained relative to the denoiser”: The training is solely led by denoising loss, specified in line 196 - 201. This joint training strategy is further discussed in sec 3.5, where we specified: “In Patronus, the prototype encoder is jointly optimized with the denoiser.” (line 254)
> * “whether prototypes are learned or chosen a priori”: Prototypes are learned, which is mentioned quite a lot of times in our paper. In sec 3 we mentioned, “This module learns patch-based prototypes within the image and computes similarity scores for each prototype, which are then used to condition the diffusion process”. Sec 3.3 introduces how to visualize the learned prototypes. Sec 4.1 is verifying that the learned prototypes are semantically meaningful. This is the core part of our paper, that we’d like to investigate what diffusion model learned as the visual prototypes to bring the interpretability.
>
> * And secondly, we’d like to acknowledge that we can see how that information does not come as how reviewer JUEx expected, and we really appreciate this perspective. We would like to specify the structure of the prototype encoder, and make the training strategies a solo subsection to make it more clear in the revised paper. We can provided moreinformation(also in the appendix) of the implementation details, and we will also release the code with the paper to ensure the reproductivity.
>
> * “Cryptic way”(also for Q4): It would be appreciated if the reviewer could give a bit more details about this part. For example, which paragraph makes you feel like that. We will do our best to increase the paper clarity. And we would also like to explain any confusions you might further have.
>
> **(2) Name inconsistency (--> W2)**
>
> Yes, you're absolutely right, and we’re aware of this.
>
> When it comes to naming models or frameworks in the ML community, there really aren’t clear rules. Some papers even introduce names without any proper long-form version. For instance, [1] presents a framework named George, which doesn’t stand for anything longer throughout the whole paper.
> Since this is a very minor concern, we’d like to keep our short name if possible.
> However *we would be willing to revise this if there is consensus between reviewers*.
>
> [1] Sohoni, Nimit, Jared Dunnmon, Geoffrey Angus, Albert Gu, and Christopher Ré. "No subclass left behind: Fine-grained robustness in coarse-grained classification problems." Advances in Neural Information Processing Systems 33 (2020): 19339-19352.
>
> **(3) Proposition lacks rigor (-->W3)**
>
> We improved the proposition and proof in the manuscript accordingly, please check the new sec 3.5. We hope this is more clear compared to the previous version.
>
>
> **(4) Reverse DDIM reference is vague (--> W4&Q3)**
>
> Reviewer JUEx mentioned that it is not clear how reverse DDIM is defined and how it is implemented in this work. We DO have this part in our paper. The details of how to implement reverse DDIM is in sec 3.3, under the paragraph called “Deterministic reverse process via DDIM sampling.” (line 234 - 245)

---

> ### Author Response · Authors · 2025-11-20
> **Reply to Reviewer JUEx [(5)Scalability & (6)quantitative evaluation over the interpretability & (7) Reply to Q2]**
>
> **(5) Scalability to higher-resolution (-->W5)**
>
> Apart from standard datasets, we have also tested on CheXpert dataset with the resolution of 224x224, which serves as evidence that our method could scale to higher resolutions and complex semantics.
>
> We would like to emphasise that:
> * The use of standard benchmark datasets is appropriate for a methodology paper. Our evaluation follows the protocol used in the most relevant work (DiffAE, InfoDiff), therefore out setup is aligned with established practice.
> * Importantly, our method is the FIRST among interpretable diffusion to additionally incorporate substantially more challenging datasets, i.e. medical imaging, providing direct evidence of scalability beyond prior work.
> * While broader datasets could further strengthen the empirical section, this is outside the scope of a paper focused on the mechanism of a prototype-based interpretable diffusion model rather than delivering an exhaustive empirical benchmark.
>
> **(6) More quantitative evaluation over the interpretability (--> W6)**
> Upon Reviwer JUEx’s request, we hereby added more quantitative evaluation of the interpretability:
>
> **a. Evaluation of interpretability quantity:**
>
> * We have sec 4.2 to examine prototype quality in a qualitative way, which examines how well our learned prototypes could estimate certain attributes, and how the learned prototypes are disentangled. This is related to what the reviewer asked for “evaluation of interpretability quality”.
> * We did an extra experiment on exploring whether we can use languages to prove the alignments of the visual concepts and human understandings. Specifically, we applied the BLIP model (Salesforce/blip-image-captioning-large in huggingface) to caption the enhanced image by prototype j and the original image, then we compared which word has a significant increase. For example, for the prototype which represents “curly hair” by visualization (Fig.3a), the top 3 words with increasement are: {‘curly’: 409, ‘hair’:307, ‘long’: 66}, the value is the word frequency with enhanced p_j minus the word frequency for the original image in 672 samples. We also provide the whole list of the top 5 increased words together with the visualization of the prototypes of one run in Appendix F.1 (Fig.17-18).
>
> **b. Faithfulness:**
>
> “faithfulness” refers to how accurately it reflects the true reasoning process of the model [1,2].
>
> We did an extra experiment to measure the faithfulness of our model, by validating whether the p score increases when enhancing it with the condition. The results are shown in the following figure.
> As we can not attach figures here, we attached the result in appendix (Fig.19).
> On x-axis, we enhanced the prototype p_i and generate a new image, on y-axis we compute the p score increasement on all prototypes, we can see from this figure that almost all prototypes are significantly increased compared to other prototypes (which basically remains unchanged with the mean ~0.0) after the enhancement, which proved that the similarity score is functioning in the way that it suppose to do.
> The 2 outliers are also interesting to see. After investigating, we believe that those 2 prototypes encode very little semantic information, from their visualization.
>
> [1]Wiegreffe, Sarah, and Yuval Pinter. "Attention is not not explanation." arXiv preprint arXiv:1908.04626 (2019).
>
> [2]Herman, Bernease. "The promise and peril of human evaluation for model interpretability." arXiv preprint arXiv:1711.07414 (2017).
>
> **c. Prototype visualization consistency (quantitative):**
>
> Upon Reviewer’s request, we quantitatively evaluated the consistency of prototype visualizations.
> Because our visual concepts are very small patches, we measure consistency directly in pixel space, which faithfully captures their low-level structural differences.
>
> Using Euclidean (L2) distance between visualization pairs, we observe a clear separation: within-prototype = 2.31 vs. between-prototype = 3.95. A two-sample t-test confirms this separation is highly significant (t = 19.55, p = $1.6×10^{-37}$).
> This confirms that each prototype forms a coherent and distinguishable visual concept.
>
> **(7) Q2: Are prototypes randomly initialized and learned, or do you initialize from real patches?**
>
> They are completely randomized and learned during the training. We intentionally avoid hard-coding semantics, leaving prototype learning unconstrained, allowing most contributing concepts to emerge with optimization. Patronus lets us examine what has been learned; this transparency enables diagnosis of entangled (Fig. 15a, appendix), biased (Fig. 4a), or missing concepts (lines 465-468), which remain hidden in standard diffusion models.
>
> --------
>
> **We thank again for Reviewer JUEx's feedback. Please reply back if there’s any further questions regarding our reply!**

---

> > ### Comment · Reviewer_JUEx · 2025-11-26
> > **Thank you for the reply**
> >
> > I thank the authors for replying to my concerns. I realized I misread a few steps throughout the paper, which you clarified in your reply. I will raise my score to 6.
> > Also, while I agree with your observation of the naming saying there are "no clear rules", I believe it's weird to use a name which looks like an acronym (underlining the "acronym" when you present the name for the first time), when the underlined letters does not align with the name of the method itself. At this point, it would have been better to just use a name with no explicit reference to any acronym.
> > Clearly, this is not a scientific limitation, therefore I am not asking you to change it, I just want to point this out.

---

> > > ### Author Response · Authors · 2025-11-26
> > > **Reply to Reviewer JUEx: thank you for your positive feedback!**
> > >
> > > We thank reviewer JUEx for your response, and we are very glad to hear that **our reply solved your concern**!!
> > > Thank you again for your valuable feedback, and for increasing the score!
> > >
> > > About the names of the model -- we agree that underlining the acronym may be unnecessary, and we will remove it in the next revision. Also, if this issue is raised again by other reviewers, we will consider changing the name to something more general, such as DiffProto.

---

### Author Response · Authors · 2025-12-02
**Summary and our message to AC and our reviewers**

Here is our message for our AC and for our reviewers:
# To AC
Thank you for your time and contribution for the reviewing process, especially after the sudden chaos. We summarize (1) paper contribution and strength; (2) what happened during the discussion phases (until the forum was frozen); (3) Changes we implemented in response to the reviewers’ feedback.

## Paper contributions and strengths
1. **Interpretability + Diffusion**:  We proposed *an intrinsic interpretable diffusion model* by integrating a prototype module for semantic encoding in visual patches.
2. **How to interpret? By visual patches (what/when/where)**: Our model gives a novel type of interpretation by revealing what visual patterns are learned and where and when they emerge throughout denoising.
3. **Solid validation**: We validated the proposed methods in 4 datasets, including CelebA (human face) as well as more complicated datasets like chest x-rays, demonstrating both faithful interpretability and strong generative performance.
4. **Detect Shortcut learning**: We also showed that our model can help in diagnosing bias and unwanted correlation and enabling the detection of shortcut learning.

## Activities during discussion phase (from 12th Nov, til 27th Nov)
1. We replied to all reviewers upon their concerns on *20th Nov*.
2. One of our reviewers, JUEx, acknowledged that our replies solved their concern and **raised the score from 4 to 6**.
3. One of our reviewers, hbds, had multiple rounds of conversation with us, and they commented that our replies “alleviated my concerns”, and “I **will increase my rating** when it is possible to edit scores”. But hbds is not able to do that because of the accident.
4. It is a pity that the other two reviewers did not get the chance to reply to us on whether our response solved their concern, due to the accident.


## Changes we made based on reviewers’ feedback
1. **Additional quantitative evaluation of interpretability**

Reviewers asked for more quantitative evaluation for the interpretability. We added two more experiments.

a. *Language alignment of the learned prototypes*: we implemented a captioning model to describe the learned prototypes, showing that the semantics of learned prototypes are aligned with human language.

b. *Faithfulness of the prototype condition*: we measure whether the score of prototype p_i increases when the corresponding condition is enhanced.

(Details in replies to JUEx.6/WfoU.2/o2Si.3/Appendix F)

2.  **Robustness of the experiments**

a. *Prototype consistency across run*: we analyse this by comparing prototype activation patterns on the same input batches of random runs. (Details in replies to hbds.5))

b. *Consistency on prototype visualization*: we measure consistency directly in pixel space with L2 distance (Details in replies to JUEx.6c)

c. *Robustness of prototype condition control*: we assessed the robustness of prototype activation control by comparing the cosine similarity between clean images and images generated with noise-perturbed prototype activations (Details in reply to hbds round2 - Q3, appendix H)

3. **Analysis on interpretability vs. performance trade-off**

We analyze the computational cost during both training and inference, showing that the latency introduced by the prototype module is minimal. We provide supporting evidence from both empirical measurements (e.g., wall-clock time and FID score) and theoretical complexity metrics (e.g., GFLOPs and number of parameters).  (Details in reply to o2Si.1)


**We thank AC for your valuable time!**

------
# To our reviewers
It is really a pity that the discussion phase ended earlier due to the accident; we could have had more interesting conversations. We appreciate your time and effort in this reviewing process. And we would like to promise you that, we have not accessed, and will not access, any leaked data containing personal information.

We know this has been a confusing and chaotic time for the whole community, and we sincerely hope that all of you are doing okay.

We thank you again! And hope you all take care!

---

### Meta-Review · Area_Chair_QLMn · 2026-01-06

**Summary:**

The authors propose an interpretable diffusion model that integrates a prototypical network to achieve intrinsic transparency by learning patch-based semantic prototypes. The authors claim that unlike post-hoc interpretation methods, Patronus reveals "what" visual patterns are learned, "where" they are located spatially, and "when" they emerge temporally during the denoising process. By using prototype activation vectors to condition the diffusion process, the model enables controllable generation and the diagnosis of model biases, specifically demonstrating the ability to detect shortcut learning such as spurious correlations. The framework is evaluated on various image datasets.

The reviewers liked the "what-where-when" interpretability framework. Reviewers also praised the model's ability to diagnose shortcut learning and undesired correlations which may have potential impact for safety and fairness. However, the initial submission faced significant criticism regarding the clarity of the methodology (e.g. description of the encoder and the lack of rigorous definitions for the reverse DDIM process). Other technical weaknesses raised were  (1) absence of quantitative metrics, (2) concerns about the computational cost of the prototype module, and (3) skepticism regarding the method's scalability to high-complexity datasets. I think overall given the indications given by the reviewers 3 out of 4 would have raised their score. My own reading is also despite the flaws the paper is in an interesting direction and merits publications.

**Reviewer Concerns:**

The authors effectively addressed concerns regarding the lack of quantitative evaluation by introducing new experiments measuring language alignment. They also providing a detailed Analysis on interpretability vs. performance trade-off. The scalability issue to more complex data and novelty concerns was not as well addressed.

**Reviewer Scores:**

Reviewer JUEx: Score increased from 4 to 6 during the discussion.
Reviewer hbds: Score likely to increase to 7. The reviewer explicitly stated, "I will increase my rating when it is possible to edit scores".
Reviewer WfoU: Score unlikely to increase. While authors addressed quantitative concerns did not clarify novelty as well and this reviewer things novelty is marginal.
Reviewer o2Si: Score likely to increase to 5. The authors provided the interpretability vs. performance experiments.

---

### Decision · Program_Chairs · 2026-01-26

Accept (Poster)